# TimePoint: Accelerated Time Series Alignment via Self-Supervised Keypoint and Descriptor Learning

**Ron Shapira Weber** [1 2] **Shahar Ben Ishay** [1 2] **Andrey Lavrinenko** [1 2] **Shahaf E. Finder** [1 2] **Oren Freifeld** [1 2 3]

## Abstract

Fast and scalable alignment of time series is a fundamental challenge in many domains. The standard solution, *Dynamic Time Warping (DTW)*, struggles with poor scalability and sensitivity to noise. We introduce *TimePoint*, a self-supervised method that dramatically accelerates DTW-based alignment while typically improving alignment accuracy by learning keypoints and descriptors from *synthetic data*. Inspired by 2D keypoint detection but carefully adapted to the unique challenges of 1D signals, TimePoint leverages *efficient 1D diffeomorphisms*—which effectively model nonlinear time warping—to generate realistic training data. This approach, along with fully convolutional and wavelet convolutional architectures, enables the extraction of informative keypoints and descriptors. Applying DTW to these sparse representations yields *major speedups* and typically *higher alignment accuracy* than standard DTW applied to the full signals. TimePoint demonstrates strong generalization to real-world time series when trained solely on synthetic data, and further improves with fine-tuning on real data. Extensive experiments demonstrate that TimePoint consistently achieves faster and more accurate alignments than standard DTW, making it a scalable solution for time-series analysis. Our code is available at https://github.com/BGU-CS-VIL/TimePoint.

## 1. Introduction

Time series data are ubiquitous across finance, healthcare, environmental monitoring, and engineering. They consist of

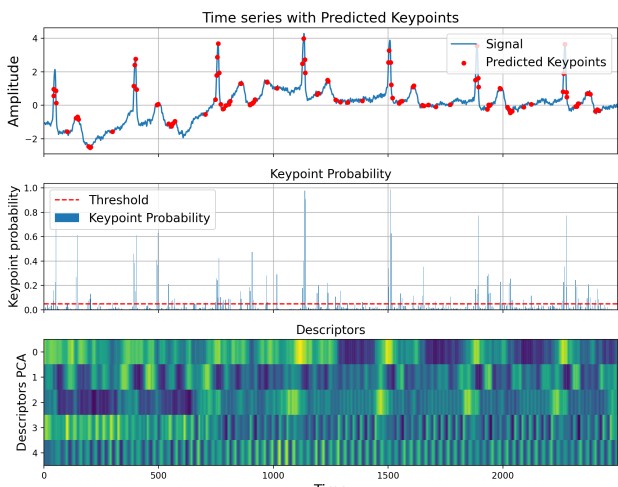

*Figure 1.* TimePoint (TP): Keypoint Detection and Descriptors on real-world, unseen, ECG data of length 2500 (TP was trained on synthetic data of length 512). Each panel depicts (top-to-bottom) the original signal and predicted keypoints, keypoint probability map, and PCA of the learned descriptors ($D = 256$, using 5 principal components for visualization purposes).

ordered sequences of observations collected over time and are instrumental in revealing temporal patterns, trends, and anomalies. A key challenge arises when these sequences grow in length or vary in sampling rates: not only do computational costs escalate with sequence size, but subtle misalignments can arise due to variable speeds or temporal distortions in the data. Consequently, developing robust and efficient approaches for comparing and aligning time series has become essential for tasks such as classification, clustering, and anomaly detection.

*Dynamic Time Warping (DTW)* is one of the most widely used algorithms for time series alignment due to its ability to accommodate elastic shifts in the temporal axis (Sakoe, 1971; Sakoe & Chiba, 1978). It has found broad application in speech recognition, gesture analysis, and signature verification. Despite its flexibility, DTW suffers from several practical limitations. First, its computational cost is $\mathcal{O}(L^2)$ with respect to the sequence lengths $L$, which becomes prohibitive for large-scale or high-throughput applications.

---

*Equal contribution [1]Department of Computer Science, Ben-Gurion University of the Negev (BGU). [2]Data Science Research Center, BGU. [3]School of Brain sciences and Cognition, BGU. Correspondence to: Ron Shapira Weber <ronsha@post.bgu.ac.il>.

*Proceedings of the $42^{nd}$ International Conference on Machine Learning*, Vancouver, Canada. PMLR 267, 2025. Copyright 2025 by the author(s).

Second, DTW is sensitive to noise or abrupt time distortions, potentially yielding suboptimal alignments under significant temporal variations or amplitude noise.

In computer vision, keypoint (KP) detection and description techniques, such as SIFT (Lowe, 1999) and Super-Point (DeTone et al., 2018), are foundational for tasks like image registration, object recognition, and 3D reconstruction. These methods identify KPs and compute corresponding descriptors that facilitate robust matching. Super-Point (SP), in particular, leverages a self-supervised learning scheme to jointly detect KPs and generate discriminative descriptors. Due to a limited large-scale dataset with known KPs, SP is first pre-trained on a synthetic dataset and later fine-tuned on real-world data.

However, an equivalent methodology for one-dimensional (1D) time series data remains largely unexplored. This gap arises from several unique challenges: time series often exhibit more complex and variable transformations (more than, e.g., homographies in images), suitable synthetic data with labeled KPs are not readily available, and existing models must handle arbitrarily long sequences without exploding in parameter count or computational cost.

We introduce *TimePoint*, a self-supervised method for KP detection and description in time series. Inspired by SuperPoint (DeTone et al., 2018), we adapt KP learning to 1D signals by modeling nonlinear temporal misalignments using Continuous Piecewise Affine Based (CPAB) transformations (Freifeld et al., 2017). To enable training, we generate synthetic time series with known KPs and apply CPAB warps to produce paired examples with ground-truth correspondences. This framework also supports fine-tuning on real-world data without architectural changes. Once trained, *TimePoint* extracts sparse KPs and descriptors that allow DTW to operate on salient locations only, significantly reducing runtime while improving robustness to noise. The model architecture is fully convolutional, combining standard and wavelet-based convolutions (Finder et al., 2024), and scales efficiently to variable-length inputs, unlike transformer-based methods, which incur quadratic cost in sequence length. Figure 1 shows *TimePoint* applied to a long ECG signal ($L = 2500$) (Dau et al., 2019). Despite being trained solely on synthetic data of length $L = 512$, *TimePoint* accurately detects salient locations and computes meaningful descriptors. To visualize these 256-dimensional features, we reduce the dimension via PCA to 5, providing an interpretable representation of the learned embeddings.

**Our contributions are as follows:**

- **A 1D Keypoint Detection and Description Framework:** We introduce *TimePoint*, a self-supervised KP detection and description method for time series data.

- **a Synthetic Dataset for Time Series Alignment:** We

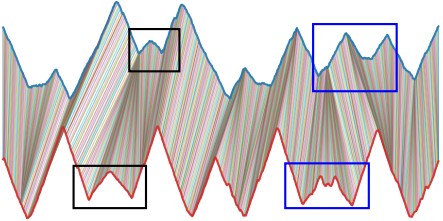

(a) DTW alignment path on dense data

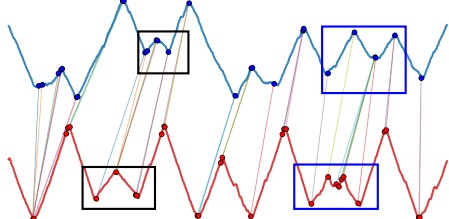

(b) DTW alignment on sparse TP features

*Figure 2.* Comparison of DTW alignment using the raw sequence (top) or TimePoint keypoints and descriptors (Bottom). The black and blue boxes highlight areas where sparse DTW using TP descriptors results in better matching.

design a synthetic time series dataset (SynthAlign) with known KPs and apply CPAB warps to generate training pairs with ground-truth correspondences.

- **Efficient Multiscale Network Architectures:** We adapt the recently proposed WTConv architecture for 1D signals, maintaining a constant parameter count regardless of the sequence length, enabling fast, scalable training and inference.

- **Fast and Sparse DTW Alignment:** We demonstrate that DTW performed on learned keypoints and descriptors yields more accurate and efficient alignment at a lower computational cost.

## 2. Related Work

**Dynamic Time Warping (DTW)** was introduced by Sakoe (Sakoe, 1971) and later refined in the seminal work by Sakoe and Chiba (Sakoe & Chiba, 1978). It is a classic method for aligning time series that differ in speed or phase by nonlinearly warping the temporal axis. Despite its widespread usage, DTW suffers from a quadratic time and memory complexity, making it challenging to apply at scale. Moreover, DTW can be overly sensitive to noise, as it attempts to align every point in both sequences. These limitations underscore the need for methods that reduce computational complexity without sacrificing accuracy, such as the sparse KP-based alignment framework we propose in

this paper. Figure 2 illustrates the difference between dense DTW alignment and the sparse KP matching approach introduced by *TimePoint*.

Several variants aim to mitigate DTW's limitations. **Soft-DTW** (Cuturi & Blondel, 2017) replaces the hard-minimum operator with a differentiable approximation, facilitating gradient-based optimization. However, it retains the core quadratic complexity in both computational time and memory due to the large cost matrices and gradients. **ShapeDTW** (Zhao & Itti, 2018) incorporates local shape descriptors into the alignment process, yielding better matching of local patterns but introducing additional computational overhead. **DTW with Global Invariances (GI-DTW)** (Vayer et al., 2020) handles global scaling and offset differences, yet remains bounded by a quadratic cost. **Fast-DTW** (Salvador & Chan, 2007) approximates the standard DTW algorithm in linear time at the cost of some accuracy; however, it was shown that under some assumptions, the speed-up could be marginal (Wu & Keogh, 2020). While we do not explore this in our experiments, it is worth noting that the use of TimePoint's KPs and descriptors could potentially accelerate FastDTW further by operating on a sparse representation of the data, similarly to how TimePoint accelerates DTW by focusing alignment on a sparse subset of KPs rather than the entire sequence.

In contrast to all these methods, **TimePoint** explicitly detects KPs and descriptors, enabling reduced dimensionality for the alignment problem while achieving accurate and scalable performance.

**CPAB Transformations for modeling time warping.** Modeling the nonlinear temporal distortions that time series often exhibit is a non-trivial task. Unlike image pairs, which can often be modeled via homographies, there is no gold-standard transformation family for time series. While it is well understood that diffeomorphisms are a natural choice to model time warping (Mumford & Desolneux, 2010), the associated computational difficulties historically hindered this approach. Fortunately, Continuous Piecewise Affine Based (CPAB) transformations (Freifeld et al., 2015; 2017) provide a flexible and efficient way to parameterize diffeomorphisms. In 1D, CPAB transformations were used in deep learning for constructing activation functions (Chelly et al., 2024; Mantri et al., 2024), multi-task fine-tuning (Mantri et al., 2025), and, most relevant in our context, modeling time warping (Weber et al., 2019; Martinez et al., 2022; Weber & Freifeld, 2023). For example, the Diffeomorphic Temporal Alignment Network (DTAN) (Weber et al., 2019) used CPAB for weakly supervised time series averaging. However, DTAN does not provide descriptors or KP detection for its inputs and requires class labels during training.

**Deep Learning for Time Series Alignment.** In the aforementioned DTAN and its variants (Weber et al., 2019; Kauf-

man et al., 2021; Martinez et al., 2022; Weber & Freifeld, 2023; 2025), the goal is to predict CPAB warps for time series averaging. TAP (Su & Wen, 2022) aims to predict the optimal alignment between time series pairs in a supervised manner but ignores the order-preserving quality of most alignment algorithms. Deep declarative DTW (Xu et al., 2023) predicts the alignment in an end-to-end manner but with increased complexity. Warpformer (Zhang et al., 2023) generates alignment paths between irregularly sampled time series. In the domain of **few-shot action recognition**, key works such as DeepCTW (Trigeorgis et al., 2016), OTAM (Cao et al., 2020), TTAN (Li et al., 2022), and TCCL (Dwibedi et al., 2019) focus on learning representations from videos using temporal alignment. However, these methods do not produce descriptors or detect KPs.

**Summary.** In contrast to existing work, **TimePoint** explicitly detects KPs and learns discriminative descriptors for alignment *using only synthetic data* (while results even further improve upon fine tuning on real data). Unlike traditional methods that are computationally expensive or deep learning approaches that rely on approximations or supervision, TimePoint restricts DTW to a small set of KPs and meaningful descriptors, significantly reducing computational overhead while often improving alignment accuracy.

## 3. SynthAlign: Synthetic Data Generation for Keypoints Detection and Matching

In this section we present `SynthAlign`, a synthetic time series and KP generator designed with the goal of facilitating self-supervised KPs detection and descriptors learning (see also Figure 3, left). This includes the data generation, KP annotation process, and augmentation strategies.

### 3.1. Challenges of Keypoint Detection in 1D Signals

While **TimePoint** draws inspiration from the 2D KP detector **SuperPoint** (DeTone et al., 2018), adapting its framework to 1D signals introduces unique challenges. First, in 2D images, KP detection and description often leverage well-defined local patches and a low-dimensional transformation family (e.g., homographies). Time series, however, frequently exhibit significant *nonlinear distortions*, including varying speeds and local stretching or compression, which cannot be well approximated by a low-dimensional transformation family. Second, *amplitude variations*, caused by noise or changes in sampling rates, can obscure salient events and complicate the task of identifying KPs. Taken together, these challenges complicate the creation of synthetic data of 1D signals, for training a model for KP detection and description (note this is a different task from generating 1D synthetic data for, say, training forecasting models).

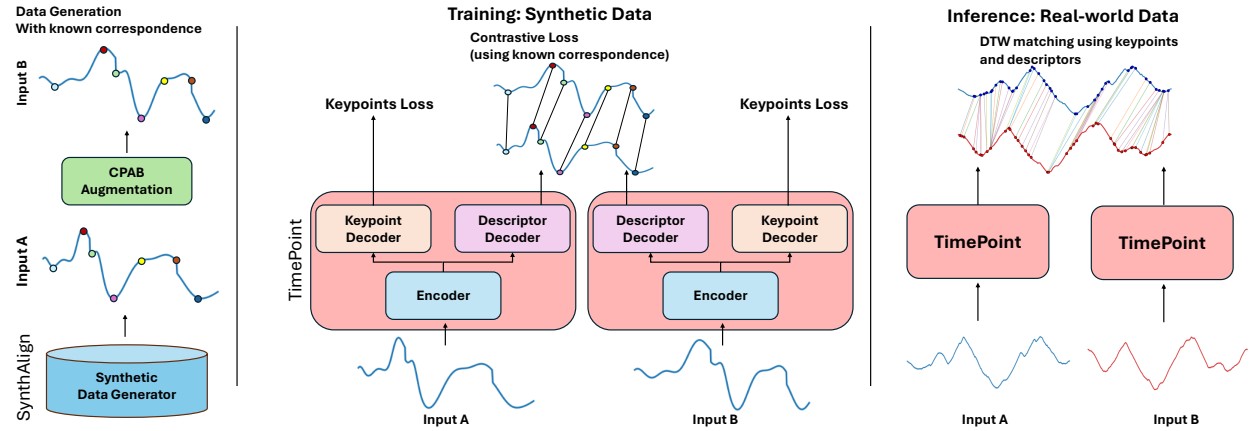

*Figure 3.* Training and Inference overview. Left: signals and keypoints are synthetically generated and augmented using CPAB warps (Section 3). Middle: *TimePoint* predicts KP location and descriptors using the known correspondence (Section 4). Right: real-world, unseen data pairs are matched using DTW on *TimePoint* descriptors at keypoint locations.

## 3.2. Generation of Synthetic Data

We generate synthetic signals with known KPs and augment them such that the correspondences between original and augmented signal pairs are known (see subsection 3.3). These synthetic data pairs provide a controlled environment for the model to learn fundamental temporal features (descriptors) and KP detection. We generate a large-scale synthetic dataset, named SynthAlign, by composing patterns and trends from a pre-defined bank of signal types. Unlike previous works in time series representation learning that synthesize signals for data augmentation (Fu et al., 2024; Ansari et al., 2024), our approach focuses specifically on generating patterns that are useful for time series alignment and also includes explicit KP generation. SynthAlign consists of the following pattern types:

- **Sine Wave Composition:** A combination of sine waves with varying frequencies and amplitudes to simulate oscillatory patterns.

- **Block, Triangle, and Sawtooth Waves:** Signals with square, triangular, or sawtooth waveforms to represent abrupt changes or linear ramps.

- **Radial Basis Functions (RBF):** Mixtures of Gaussian blobs to model localized smooth events.

KPs for each pattern are derived from salient features, including pattern start and end points, peaks, and derivative zero crossings. We denote $X \in \mathbb{R}^L$ and $Y = (y_t)_{t=1}^L$, with $y_t \in \{0, 1\}$, as the synthetic signal and its KPs respectively (where $L = 512$). To further diversify the data, we superimpose linear trends to emulate non-stationary behavior, randomly flip sections or entire signals, and add Gaussian noise (jitter) $\epsilon \sim \mathcal{N}(0, 0.1)$. This suite of pattern composition and

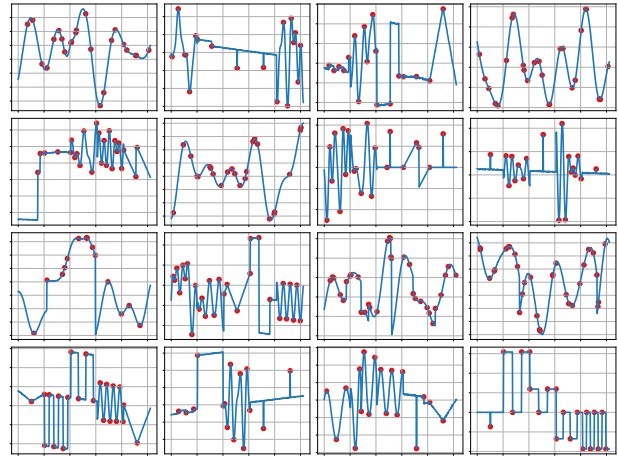

*Figure 4.* Samples from the synthetic dataset SynthAlign.

augmentations allows the model to handle real-world variations effectively. Samples from SynthAlign are shown in Figure 4 (see more details in Appendix B).

## 3.3. Generation of Correspondences Using CPAB Warps

As stated in (DeTone et al., 2018), *"a homography is a good model for what happens when the same 3D point is seen from different viewpoints"*. Adopting a similar approach to time series, requires a family of transformations, in 1D, that provide a good model for nonlinear time warping. With this in mind, we use the CPAB transformations (Freifeld et al., 2017) to simulate nonlinear temporal distortions and generate correspondences between original and transformed signals. CPAB transformations are parametric, highly-expressive, and computationally-efficient diffeomorphisms (a diffeomorphism is an invertible map with a

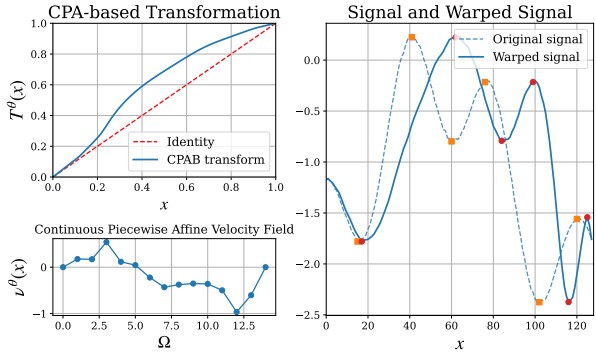

*Figure 5.* Generating signals and keypoints pairs with known correspondences using a CPAB transformation $T^{\boldsymbol{\theta}}$, which was obtained from a CPA velocity field, $\boldsymbol{v}^{\boldsymbol{\theta}}$, as proposed in (Freifeld et al., 2017) and briefly explained in our Appendix.

differentiable inverse). Briefly, a CPAB transformation is obtained by integrating a Continuous Piecewise Affine (CPA) velocity field. The term "Piecewise Affine" is w.r.t. some partition of the domain (as shown in Figure 5). While the full details behind CPAB transformations – which are available in (Freifeld et al., 2017) – are inessential for understanding our paper, Appendix C contains the key details of how a CPAB transformation is defined and built.

Given an original signal $X$, we generate a warped signal $X' = X \circ T^{\boldsymbol{\theta}}$, where $T^{\boldsymbol{\theta}}$ is a CPAB transformation parameterized by $\boldsymbol{\theta}$. The known transformation provides ground truth correspondences between $X$ and $X'$, enabling supervised learning for KP detection and descriptor matching.

Not all diffeomorphic transformations yield realistic temporal distortions. To ensure plausible time warping, we sample $\boldsymbol{\theta}$ from $\mathcal{N}(\mathbf{0}, \boldsymbol{\Sigma}_{\text{CPA}})$ the zero-mean Gaussian smoothness prior from (Freifeld et al., 2017). That prior, over CPA velocity fields, penalizes large or abrupt deformations. Its covariance matrix, $\boldsymbol{\Sigma}_{\text{CPA}}$, has two hyper-parameters, $\sigma_{\text{var}}$ and $\sigma_{\text{smooth}}$, which govern the variance and smoothness, respectively, of the CPA velocity field, hence also of the resulting CPAB transformation. Setting $\sigma_{\text{smooth}} = 1$ and $\sigma_{\text{var}} = 0.5$ imposes mild constraints on the velocity fields, favoring nearly affine local segments without overly restricting their overall variability. In our experiments, we partition the domain into 16 segments (so, due to zero-boundary conditions, $\dim(\boldsymbol{\theta}) = 15$; see (Freifeld et al., 2017)) to balance flexibility and simplicity in warping.

# 4. TimePoint Architecture

We now describe TimePoint's overall architecture and its components, followed by the loss functions. The entire training and inference pipeline is illustrated in Figure 3. The architecture, detailed Figure 6, consists of a shared encoder

and two decoders: one specialized for KP detection and the other for descriptor computation. By leveraging a fully convolutional network with Wavelet Transform Convolution (WTConv) layers, TimePoint efficiently captures multiscale temporal features. This design maintains a fixed number of parameters, regardless of the input's length, $L$, ensuring scalability to long sequences.

## 4.1. Shared Encoder

The shared encoder processes the input time series $X \in \mathbb{R}^{C \times L}$, where $C$ is the number of channels and $L$ is the signal length. In this work, we focus on the univariate case ($C = 1$), with multivariate data left for future exploration. We employ the recently proposed **WTConv layer** (Finder et al., 2024), which operates in the wavelet domain to capture patterns at multiple scales. Using a 3-level wavelet decomposition with kernel size 3, the WTConv layers efficiently learn both low-frequency (global) and high-frequency (local) features. Each WTConv layer is followed by batch normalization. A stride of 2 is applied between each of the 3 WTConv blocks, resulting in an overall downsampling factor of 8.

The encoder produces a feature map $F \in \mathbb{R}^{D_{\text{enc}} \times L'}$, where $D_{\text{enc}}$ is the feature dimension, and $L'$ represents the length after downsampling. This feature map forms the basis for subsequent KP detection and descriptor generation.

## 4.2. Keypoint Decoder

The keypoint decoder processes the feature map $F \in \mathbb{R}^{D_{\text{enc}}}$ ($D_{\text{enc}} = 256$ in our experiments) produced by the shared encoder to predict keypoint scores for each temporal location. It consists of a convolutional layer that refines the features for KP detection and maps $\mathbb{R}^{D_{\text{enc}} \times L'} \mapsto \mathbb{R}^{8 \times L'}$ (analogous to the 'cell size' of size 8 in (DeTone et al., 2018)). The cells are then reshaped from $8 \times L'$ back to $L$ followed by a sigmoid activation function. The output is a score vector $S = (s_t)_{t=1}^{L}$, with $s_t \in [0, 1]$, where each entry represents the probability of a KP at time step $t$. Non-Maximum Suppression (NMS) is then applied with a window size of 5 to suppress redundant detections. KPs are selected by either applying a pre-defined threshold or choosing the top-$K$ timesteps with the highest probability.

## 4.3. Descriptor Decoder

The descriptor decoder also operates on $F$ but is tasked with generating a descriptor vector for each time step. A convolutional layer first maps $F$ into a descriptor space ($D_{\text{desc}} = 256$), after which an upsampling operator restores the temporal dimension from $L'$ back to $L$. We then apply $\ell_2$ normalization so that each descriptor lies on the unit hypersphere. The resulting output, $F_{\text{desc}} \in \mathbb{R}^{D_{\text{desc}} \times L}$, serves

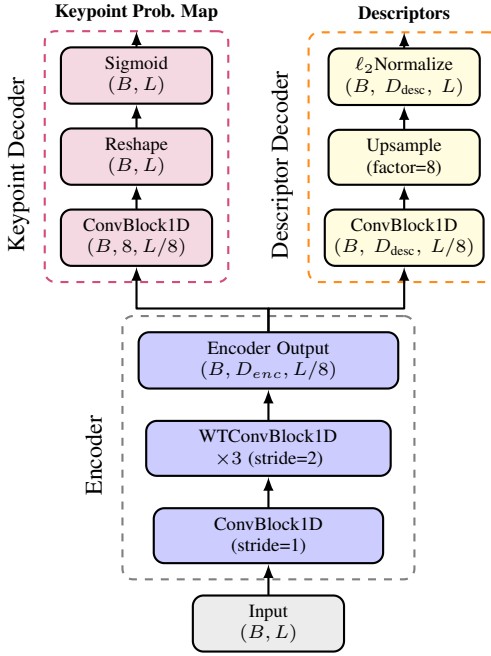

*Figure 6.* TimePoint model architecture.

as a dense descriptor matrix from which we extract a sparse set of descriptors at the KP locations identified by the decoder above. This results in a compact representation for alignment tasks.

## 4.4. Loss Functions

We train **TimePoint** in a self-supervised manner by sampling synthetic signals and KPs $(X, Y)$ from `SynthAlign` (as described in Section 3), where $X$ is a time series and $Y$ its associated keypoint labels. To simulate realistic temporal distortions, we generate $(X', Y')$ via a known CPAB transformation $T^{\theta}$ (having sampled $\theta$ from the prior), i.e. $X' = X \circ T^{\theta}$ and $Y' = Y \circ T^{\theta}$. This process ensures we have warped signals pairs with ground-truth correspondences, facilitating KP detection and descriptor learning.

**Keypoint Detection Loss.** Keypoint detection is formulated as a binary classification task at each time step. Consequently, we use a binary cross-entropy loss between the predicted scores $S = (s_t)_{t=1}^{L}$ and the ground-truth labels $Y = (y_t)_{t=1}^{L}$:

$$\mathcal{L}_{\mathrm{kp}}(S, Y) = -\frac{1}{L} \sum_{t=1}^{L} \Big[ y_t \log(s_t) + (1 - y_t) \log\big(1 - s_t\big) \Big].$$
(1)

Here, $y_t \in \{0, 1\}$ indicates whether a keypoint is present at time $t$, and $s_t$ is the predicted probability.

**Descriptor Loss.** Let $N \ll L$ be the number of ground-truth KPs from the original and warped signals, $Y$ and $Y'$. We define a set of matched indices $\mathcal{G} \subseteq \{1, \ldots, N\} \times \{1, \ldots, N\}$ where $(i, j) \in \mathcal{G}$ if and only if the $i$-th keypoint in $Y$ corresponds to the $j$-th keypoint in $Y'$ under the known transformation $T^{\theta}$. Denoting the descriptors at these KPs as $(D_i)_{i=1}^{N}$ and $(D'_j)_{j=1}^{N}$, we compute a margin-based contrastive loss only over these KP descriptors. Specifically, for each pair $(i, j)$, we treat it as a *positive* (matching) pair if $(i, j) \in \mathcal{G}$, and as a *negative* pair otherwise. The loss is then

$$\mathcal{L}_{\mathrm{desc}}(D, D') =$$
$$\frac{1}{N^2} \sum_{i=1}^{N} \sum_{j=1}^{N} \Big[ \mathbf{1}_{\mathcal{G}}((i, j)) \, \max\big(0, m_p - \cos(D_i, D'_j)\big)^2$$
$$+ \big(1 - \mathbf{1}_{\mathcal{G}}((i, j))\big) \, \max\big(0, \cos(D_i, D'_j) - m_n\big)^2 \Big] \quad (2)$$

where $\cos(D_i, D'_j) = \frac{D_i^{\top} D'_j}{\|D_i\| \|D'_j\|}$ is the cosine similarity and $\mathbf{1}_{\mathcal{G}}((i, j))$ is the indicator function; namely, $\mathbf{1}_{\mathcal{G}}((i, j)) = 1$ if $(i, j) \in \mathcal{G}$ and 0 otherwise. We set the positive margin $m_p = 1$ to push matched pairs toward maximal similarity, and a negative margin $m_n = 0.1$ to separate non-matching pairs. Since $N \ll L$, computing the descriptor loss solely at KP locations substantially reduces memory usage and focuses the training on salient regions of the signal.

**Overall Loss.** The overall loss combines the detection and descriptor losses across the original and warped signals:

$$\mathcal{L}(S, S', Y, Y', D, D') =$$
$$\underbrace{\mathcal{L}_{\mathrm{kp}}(S, Y)}_{\text{kp detection in } X} + \underbrace{\mathcal{L}_{\mathrm{kp}}(S', Y')}_{\text{kp detection in } X'} + \underbrace{\mathcal{L}_{\mathrm{desc}}(D, D')}_{\text{descriptor matching}}. \quad (3)$$

## 4.5. DTW Alignment Using TimePoint

Once *TimePoint* is trained, its learned KPs and descriptors can be used to perform alignment more efficiently. At test time, given two input signals $X \in \mathbb{R}^L$ and $X' \in \mathbb{R}^{L'}$, we apply *TimePoint* to extract their respective KPs and descriptor sequences, denoted $D \in \mathbb{R}^{\widetilde{L} \times D_{\mathrm{desc}}}$ and $D' \in \mathbb{R}^{\widetilde{L}' \times D_{\mathrm{desc}}}$, where $\widetilde{L} \ll L, \widetilde{L}' \ll L'$ are the numbers of selected KPs.

Instead of aligning the raw signals $X$ and $X'$, we perform DTW directly on the descriptor sequences $D$ and $D'$. To account for the vector nature of the descriptors, we replace the standard scalar-based Euclidean cost in DTW with a cosine-similarity-based cost:

$$\mathrm{cost}\big(D[t], D'[t']\big) = 1 - \cos\big(D[t], D'[t']\big), \quad (4)$$

where $t$ and $t'$ index the descriptors corresponding to KPs in $X$ and $X'$, respectively.

Critically, by aligning only the sparse set of KP descriptors, the computational complexity of DTW is reduced from

$O(L \cdot L')$ to $O(\widetilde{L} \cdot \widetilde{L}')$. For instance, using 10% of the original signal length yields up to a $100\times$ speedup, while often improving alignment accuracy due to the robustness of the learned features.

### 4.6. Fine-Tuning on Real-World Time Series

While `SynthAlign` enables TimePoint to learn KP detection and descriptors in a fully synthetic setting, there may still be a distribution gap when applying the model to real-world signals. To further improve generalization, we fine-tune TimePoint directly on real data from the UCR archive (Dau et al., 2019), using a similar self-supervised protocol. We simulate temporal distortions by applying two independently sampled CPAB transformations to each signal, $X$, yielding two warped views $X_1 = X \circ T^{\theta_1}$ and $X_2 = X \circ T^{\theta_2}$. Generating two augmented versions also helps to mitigate overfitting (which is not necessary when data is generated on-the-fly in `SynthAlign`). Since both originate from the same source $X$, we know the ground-truth correspondence between any point in $X_1$ and $X_2$.

As no ground-truth KPs are available in real datasets, we adopt the same heuristic strategy used in `SynthAlign`: we mark as KPs locations of local extrema (minima and maxima), derivative zero-crossings, etc. While it is less effective on real-world data, we notice that the produced KPs are akin to training with 'noisy labels', contributing to TP's generalization and robustness.

## 5. Limitations

Although our synthetic data generation facilitates KPs and descriptors learning that can be easily transferred to real-world data in many scenarios, TP's performance might be sub-optimal if signals deviate substantially from the synthetic distribution (however, this can be mitigated by fine-tuning TP). Moreover, if the underlying temporal distortions exceed the scope of the predefined CPAB prior, it might require further adjustments. Finally, our encoder downsamples each time series by a factor of 8 to enhance efficiency. While beneficial for long signals, this fixed rate might overly compress shorter sequences.

## 6. Experiments and Results

We evaluate *TimePoint* across a range of experiments that assess the computational efficiency, robustness to noise, and classification accuracy on real-world data. An ablation study further analyzes the contribution of each component.

### 6.1. Implementation Details

Our model, implemented in `PyTorch`, has a total of $\sim$200K trainable parameters. We have adopted the 2D WT-

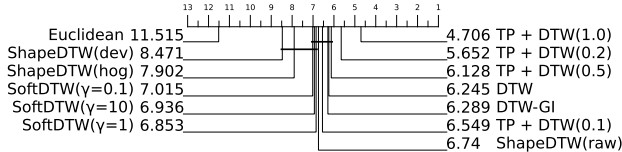

*Figure 7.* Critical Difference Diagram. The scores represent the average rank (1-NN Acc.) of each method across 102 UCR datasets. TimePoint (TP) was trained solely on synthetic data.

Conv layer from the official implementation (Finder et al., 2024) to 1D inputs. `SynthAlign` synthetic data generation occurs on-the-fly, such that no example is seen twice. Training is performed on a single NVIDIA RTX6000 GPU with 48 GB of memory. The model converges within approximately 100,000 iterations and 20 hours, with a batch size of 512, and the AdamW optimizer (Loshchilov, 2017) with a learning rate of $1 \times 10^{-4}$ with cosine learning rate scheduler. The encoder consists of 4 layers with a number of kernels $= [128, 128, 256, 256]$.

To fine-tune TP, we first train our model as detailed above on `SynthAlign`. Next, we train TP on $\sim 100$ UCR datasets for 2000 epochs. We resample each signal to $L = 512$ and follow the procedure described in subsection 4.6.

To facilitate fast evaluation of DTW $k$-Nearest Neighbors ($k$NN), we implement DTW using pytorch and perform batch-wise $k$NN. For SoftDTW (Cuturi & Blondel, 2017) we use the `CUDA` implementation[1]. Since the `CUDA` version of SoftDTW is limited to a maximum length of 1024, using fewer timesteps not only reduces computation time and RAM consumption but also allows for processing of much longer sequences by using only the selected KPs.

### 6.2. Classification on Real-World Data

TP was trained on synthetic data using the `SynthAlign` dataset (i.e., without the fine-tuning step). We evaluate its generalization to real-world data using the UCR Time Series Archive (Dau et al., 2019). The archive has 128 datasets with inter-dataset variability in the number of samples, length, application domain, and more. We use the original train-test splits provided by the archive. Thus, the results might differ from the ones reported by the original SoftDTW (Cuturi & Blondel, 2017) since they have shuffled the splits and produced new ones. We use a subset of 102 datasets, omitting ones that did not produce results for all methods (e.g., when one or more of our competitors' runtime exceeded 12 hours or due to data handling issues).

---

[1]github.com/Maghoumi/pytorch-softdtw-cuda

*Table 1.* Comparison of DTW and SoftDTW on 102 UCR datasets w/o TimePoint at various KP percentages. Top: 1-NN classification accuracy. Bottom: total runtime in GPU hours.

| Method | TimePoint 1-NN Accuracy | | | | |
|---|---|---|---|---|---|
| | Baseline | 10% | 20% | 50% | 100% |
| DTW | 0.706 | 0.707 | 0.721 | 0.710 | **0.732** |
| + fine-tuning | 0.706 | 0.777 | 0.790 | 0.769 | **0.80** |
| SoftDTW($\gamma = 0.1$) | 0.677 | 0.659 | 0.662 | 0.689 | **0.720** |
| + fine-tuning | 0.677 | 0.724 | 0.712 | 0.729 | **0.752** |
| SoftDTW($\gamma = 1$) | 0.671 | 0.654 | 0.659 | 0.687 | **0.711** |
| + fine-tuning | 0.671 | 0.72 | 0.711 | 0.722 | **0.749** |
| SoftDTW($\gamma = 10$) | 0.670 | 0.655 | 0.658 | 0.680 | **0.703** |
| + fine-tuning | 0.670 | 0.724 | 0.712 | 0.729 | **0.752** |

| Method | GPU Runtime (hours) | | | | |
|---|---|---|---|---|---|
| | Baseline | 10% | 20% | 50% | 100% |
| DTW | 192 | 0.71 | 2.88 | 19.59 | 193 |
| SoftDTW($\gamma = 0.1$) | 2.10 | 0.65 | 0.70 | 1.00 | 2.17 |
| SoftDTW($\gamma = 1$) | 1.98 | 0.52 | 0.56 | 0.86 | 2.16 |
| SoftDTW($\gamma = 10$) | 1.94 | 0.50 | 0.55 | 0.85 | 2.02 |

*Table 2.* 1-NN classification accuracy under various perturbations. Results averaged over 30 UCR datasets.

| Method | No Noise | Blur $\sigma = 0.1$ | Blur $\sigma = 1.0$ | Jitter $\sigma = 0.1$ | Jitter $\sigma = 0.5$ |
|---|---|---|---|---|---|
| DTW (raw) | 0.844 | 0.843 | 0.838 | 0.801 | 0.744 |
| TP (10%) | 0.867 | 0.866 | 0.853 | 0.804 | 0.760 |
| TP (20%) | **0.881** | **0.873** | **0.873** | **0.828** | **0.791** |

A common approach for time series classification and alignment evaluation is *k*NN with DTW as the distance measure. As baselines, we include classical DTW, SoftDTW (Cuturi & Blondel, 2017) with $\gamma \in \{0.1, 1, 10\}$, DTW-GI (Vayer et al., 2020), and ShapeDTW (using 'raw', 'derivative' and 'hog1d' descriptors) (Zhao & Itti, 2018), where we rely on the official DTW-GI implementation and `sktime` for ShapeDTW (Löning et al., 2019). For TP, we evaluate keypoint ratios w.r.t. the sequence length $\{0.1, 0.2, 0.5, 1\}$, observing that selecting fewer KPs accelerates DTW alignment while often preserving or improving classification accuracy. We select KPs by sorting the detection confidence and retaining the top $K\%$. Each value corresponds to a different threshold. This adaptive strategy avoids using a fixed threshold across all datasets, which may be sub-optimal.

We summarize performance using a critical difference diagram (Demšar, 2006; Middlehurst et al., 2024), which ranks each *k*NN classifier across multiple datasets (the full results appear in Appendix E). Classifiers are grouped into "cliques," connected by a horizontal line, if their average ranks are not significantly different according to pairwise one-sided Wilcoxon signed-rank tests with Holm correction (Middlehurst et al., 2024). As illustrated in Figure 7, *TP+DTW* achieves the highest average rank (with statistical significance) at both 100% and 20% keypoint usage, and remains highly ranked at 10% and 50%. Despite being trained exclusively on synthetic data, TP demonstrates strong zero-shot generalization to real-world time series.

### 6.3. GPU-enabled DTW

We compare TP+DTW and TP+SoftDTW to the corresponding methods using batch-wise *k*NN on the GPU. Table 1 shows the average accuracy and total runtime across the datasets. The reported runtime includes TP's forward pass, sorting KPs by probability, NMS, and DTW. Here, we also

report the results for *fine-tuning* TP. The results show that both methods enjoy significant improvement in both metrics. Comparing DTW with TP+DTW, using 20% of the KPs yields 2% accuracy gain with a $\times 65$ **speedup** (192 vs. 3 hours respectively). For SoftDTW, using 10% or 20% of the KPs is not necessarily better, since SoftDTW($\gamma$) encourages the solution to be smooth across adjacent time steps. However, using 50% of the KPs yields better accuracy (1-2%) at less than half the running time, and the full length gives a significant accuracy boost across various $\gamma$ values (4-5%) with almost identical runtimes. Finally, we provide GPU-RAM consumption analysis in subsection A.3.

To further improve performance, we fine-tune TP on real-world time series using the same self-supervised framework. Fine-tuning leads to a substantial boost in accuracy: compared to the synthetic-only model, we observe a 7–8% improvement in performance across KP ratios. Importantly, runtime remains unchanged, as the architecture and inference pipeline are identical, which demonstrates that TP can be adapted to new domains while preserving its computational advantages.

### 6.4. Runtime Analysis

We evaluate the runtime performance of DTW-*k*NN on both the raw signal and *TimePoint* descriptors with varying numbers of KPs. We compute *k*NN between two synthetic datasets, each of size $N = 500$, with sequence lengths $L \in \{50, 100, \dots, 1000\}$. The results are shown in Figure 8. When using the full signal ($L = 100\%$), the runtime for DTW and TP+DTW is almost identical. However, as the number of KPs in TP decreases, the runtime for TP+DTW scales significantly better with $L$, demonstrating near-linear behavior for long sequences. For instance, when $L = 1000$, using 20% of the KPs is *almost two orders of magnitude faster* than performing DTW on the entire length. This indicates that TP's ability to focus on a reduced set of the KPs leads to substantial efficiency gains, particularly for large-scale, long sequence datasets.

### 6.5. Robustness to Noise

To assess TP's robustness under noisy and distorted conditions, we conducted a controlled evaluation using a subset of 30 datasets from the UCR archive. We introduced two common types of perturbations: additive Gaussian noise

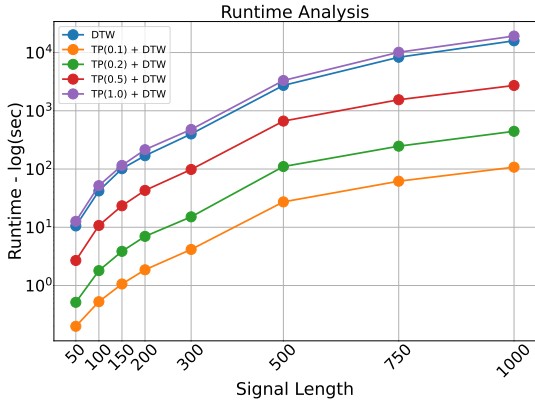

*Figure 8.* Runtime analysis. DTW KNN runtime between two synthetic datasets of $N = 500$ and varying lengths (on GPU).

*Table 3.* Ablation Study

| Encoder | Dist. | %L | NMS | 1-NN acc. |
|---|---|---|---|---|
| DTW (baseline) | Euclidean | 100% | - | 0.797 |
| TP + Dense Conv | Euclidean | 100% | - | 0.82 |
| TP + WTConv | Euclidean | 100% | - | 0.815 |
| TP + Dense Conv | Cosine Sim. | 100% | - | 0.835 |
| TP + WTConv | Cosine Sim. | 100% | - | **0.869** |
| TP + DTW (no descriptors) | Euclidean | 20% | ✓ | 0.792 |
| TP + Dense Conv | Cosine Sim. | 20% | ✓ | 0.826 |
| TP + WTConv | Cosine Sim. | 20% | ✗ | 0.841 |
| TP + WTConv | Cosine Sim. | 20% | ✓ | 0.865 |

# 7. Conclusion

We introduced TimePoint (TP), a self-supervised framework for efficient time-series alignment. By leveraging synthetic data and employing CPAB transformations, TP learns to detect KPs and descriptors that enable sparse and accurate alignments. Our approach addresses the scalability limitation of traditional methods like DTW, achieving significant computational speedups and improved alignment accuracy. Extensive experiments demonstrate that TP generalizes well across diverse real-world datasets, underscoring its effectiveness as a practical solution for time-series analysis.

# Acknowledgments

This work was supported by the Lynn and William Frankel Center at BGU CS, by the Israeli Council for Higher Education via the BGU Data Science Research Center, and by Israel Science Foundation Personal Grant #360/21. S.E.F.'s work was supported by the BGU's Hi-Tech Scholarship. S.E.F.'s and R.S.W.'s work was also supported by the Kreitman School of Advanced Graduate Studies.

# Impact Statement

This paper presents work whose goal is to advance the field of Machine Learning. There are many potential societal consequences of our work, none which we feel must be specifically highlighted here.

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

(jitter) and Gaussian blur, each applied at two intensity levels. For each condition, we repeated the experiment three times to account for randomness in the perturbations and report the average 1-NN classification accuracy using DTW on raw signals and TimePoint descriptors. We evaluate TP at two keypoint selection ratios: 10% and 20%. The results are presented at Table 2 and indicate that TP is robust to varying noise types.

## 6.6. Ablation Study

Table 3 summarizes an ablation study over 27 datasets using a 1-NN accuracy metric. With the full sequence, the standard DTW (baseline) achieves an accuracy of 0.797. Performing DTW with TP features improves performance for both Dense Conv ($\sim$400K parameters, 0.82) and WTConv ($\sim$200K parameters, 0.815) encoders when using Euclidean distance. Switching to cosine similarity further boosts performance to 0.835 for Dense Conv and 0.869 for WTConv, highlighting the effectiveness of the wavelet-based encoder. Reducing the signal to just 20% of its length yields comparable trends: first, using TP merely as subsampling (i.e., ignoring the descriptors and using DTW on the raw signals restricted to *TimePoint*'s KPs) gives a 1-NN accuracy of 0.792, while switching to using TP's descriptors, with either the Dense Conv or WTConv encoders, reaches 0.826 or 0.865, respectively. This shows that *TimePoint* differs from naive subsampling methods by not only learning an input-dependent KP detection scheme tailored to alignment but also providing descriptors that, crucially, capture nonlocal context through a large receptive field. This leads to efficient DTW alignment without sacrificing accuracy. Note also TP+WTConv nearly matches in accuracy the full-length setting even though far fewer KPs are used. Lastly, dropping the NMS decreases TP+WTConv to 0.841.

*Conference on Computer Vision*, pp. 200–217. Springer, 2024.

Cuturi, M. and Blondel, M. Soft-dtw: a differentiable loss function for time-series. In *International conference on machine learning*, pp. 894–903. PMLR, 2017.

Dau, H. A., Bagnall, A., Kamgar, K., Yeh, C.-C. M., Zhu, Y., Gharghabi, S., Ratanamahatana, C. A., and Keogh, E. The ucr time series archive. *IEEE/CAA Journal of Automatica Sinica*, 6(6):1293–1305, 2019.

Demšar, J. Statistical comparisons of classifiers over multiple data sets. *The Journal of Machine learning research*, 7:1–30, 2006.

DeTone, D., Malisiewicz, T., and Rabinovich, A. Superpoint: Self-supervised interest point detection and description. In *Proceedings of the IEEE conference on computer vision and pattern recognition workshops*, pp. 224–236, 2018.

Dwibedi, D., Aytar, Y., Tompson, J., Sermanet, P., and Zisserman, A. Temporal cycle-consistency learning. In *Proceedings of the IEEE/CVF conference on computer vision and pattern recognition*, pp. 1801–1810, 2019.

Finder, S. E., Amoyal, R., Treister, E., and Freifeld, O. Wavelet convolutions for large receptive fields. In *European Conference on Computer Vision*, pp. 363–380. Springer, 2024.

Freifeld, O., Hauberg, S., Batmanghelich, K., and Fisher III, J. W. Highly-expressive spaces of well-behaved transformations: Keeping it simple. In *ICCV*, 2015.

Freifeld, O., Hauberg, S., Batmanghelich, K., and Fisher III, J. W. Transformations based on continuous piecewise-affine velocity fields. *IEEE TPAMI*, 2017.

Fu, F., Chen, J., Zhang, J., Yang, C., Ma, L., and Yang, Y. Are synthetic time-series data really not as good as real data? *arXiv preprint arXiv:2402.00607*, 2024.

Kaufman, I., Weber, R. S., and Freifeld, O. Cyclic diffeomorphic transformer nets for contour alignment. In *2021 IEEE International Conference on Image Processing (ICIP)*, pp. 349–353. IEEE, 2021.

Li, S., Liu, H., Qian, R., Li, Y., See, J., Fei, M., Yu, X., and Lin, W. Ta2n: Two-stage action alignment network for few-shot action recognition. In *Proceedings of the AAAI Conference on Artificial Intelligence*, volume 36, pp. 1404–1411, 2022.

Löning, M., Bagnall, A., Ganesh, S., Kazakov, V., Lines, J., and Király, F. J. sktime: A unified interface for machine learning with time series. *arXiv preprint arXiv:1909.07872*, 2019.

Loshchilov, I. Decoupled weight decay regularization. *arXiv preprint arXiv:1711.05101*, 2017.

Lowe, D. G. Object recognition from local scale-invariant features. In *Proceedings of the seventh IEEE international conference on computer vision*, volume 2, pp. 1150–1157. Ieee, 1999.

Mantri, K. S. I., Wang, X., Schönlieb, C.-B., Ribeiro, B., Bevilacqua, B., and Eliasof, M. Digraf: Diffeomorphic graph-adaptive activation function. In *Advances in Neural Information Processing Systems (NeurIPS)*, 2024.

Mantri, K. S. I., Schönlieb, C.-B., Ribeiro, B., Baskin, C., and Eliasof, M. Ditask: Multi-task fine-tuning with diffeomorphic transformations. In *Proceedings of the IEEE/CVF Conference on Computer Vision and Pattern Recognition (CVPR)*, 2025.

Martinez, I., Viles, E., and Olaizola, I. G. Closed-form diffeomorphic transformations for time series alignment. In *International Conference on Machine Learning*, pp. 15122–15158. PMLR, 2022.

Middlehurst, M., Schäfer, P., and Bagnall, A. Bake off redux: a review and experimental evaluation of recent time series classification algorithms. *Data Mining and Knowledge Discovery*, pp. 1–74, 2024.

Mumford, D. and Desolneux, A. *Pattern theory: the stochastic analysis of real-world signals*. AK Peters/CRC Press, 2010.

Sakoe, H. Dynamic-programming approach to continuous speech recognition. *1971 Proc. the International Congress of Acoustics, Budapest*, 1971.

Sakoe, H. and Chiba, S. Dynamic programming algorithm optimization for spoken word recognition. *IEEE Transactions on Acoustics, Speech, and Signal Processing*, 26(1):43–49, 1978. ISSN 0096-3518. doi: 10.1109/TASSP.1978.1163055.

Salvador, S. and Chan, P. Toward accurate dynamic time warping in linear time and space. *Intelligent Data Analysis*, 11(5):561–580, 2007.

Su, B. and Wen, J.-R. Temporal alignment prediction for supervised representation learning and few-shot sequence classification. In *International Conference on Learning Representations*, 2022.

Trigeorgis, G., Nicolaou, M. A., Zafeiriou, S., and Schuller, B. W. Deep canonical time warping. In *Proceedings of the IEEE Conference on Computer Vision and Pattern Recognition*, pp. 5110–5118, 2016.

Vayer, T., Chapel, L., Courty, N., Flamary, R., Soullard, Y., and Tavenard, R. Time series alignment with global invariances. *arXiv preprint arXiv:2002.03848*, 2020.

Weber, R. S. and Freifeld, O. Regularization-free diffeomorphic temporal alignment nets. In *International Conference on Machine Learning*, pp. 30794–30826. PMLR, 2023.

Weber, R. S. and Freifeld, O. Diffeomorphic temporal alignment nets for time-series joint alignment and averaging. *arXiv preprint arXiv:2502.06591*, 2025.

Weber, R. S., Eyal, M., Skafte Detlefsen, N., Shriki, O., and Freifeld, O. Diffeomorphic temporal alignment nets. In *Advances in neural information processing systems*, volume 32, 2019.

Wu, R. and Keogh, E. J. Fastdtw is approximate and generally slower than the algorithm it approximates. *IEEE Transactions on Knowledge and Data Engineering*, 34 (8):3779–3785, 2020.

Xu, M., Garg, S., Milford, M., and Gould, S. Deep declarative dynamic time warping for end-to-end learning of alignment paths. *arXiv preprint arXiv:2303.10778*, 2023.

Zhang, J., Zheng, S., Cao, W., Bian, J., and Li, J. Warpformer: A multi-scale modeling approach for irregular clinical time series. In *Proceedings of the 29th ACM SIGKDD Conference on Knowledge Discovery and Data Mining*, pp. 3273–3285, 2023.

Zhao, J. and Itti, L. shapedtw: Shape dynamic time warping. *Pattern Recognition*, 74:171–184, 2018.

# TimePoint: Appendix

## Table of Contents

# A. Additional Results

## A.1. TimePoint Keypoints and Descriptors for Time Series of Different Lengths

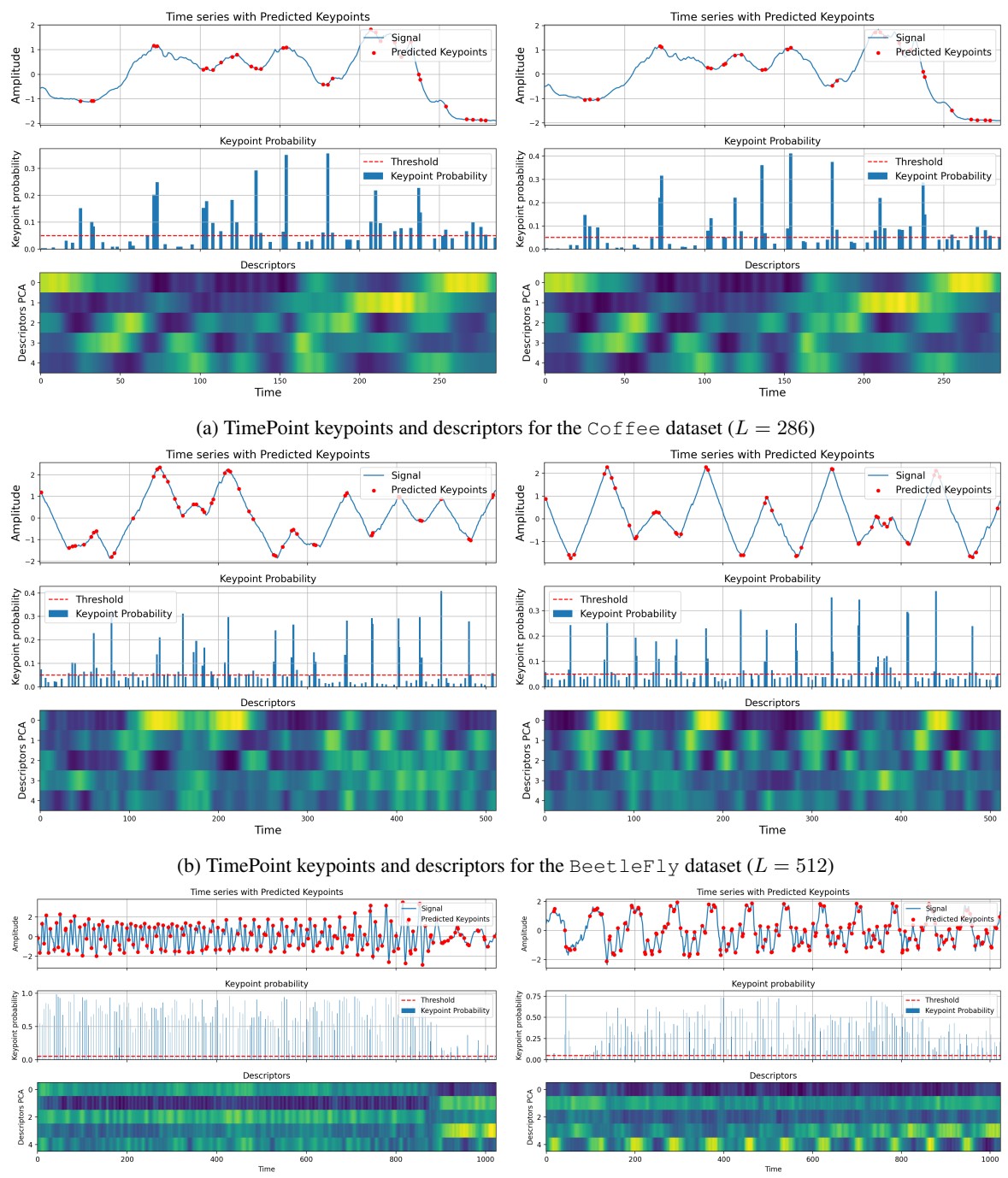

(a) TimePoint keypoints and descriptors for the `Coffee` dataset ($L = 286$)

(b) TimePoint keypoints and descriptors for the `BeetleFly` dataset ($L = 512$)

(c) TimePoint keypoints and descriptors for the `Phoneme` dataset ($L = 1024$)

## A.2. 1-NN Classification Comparisons

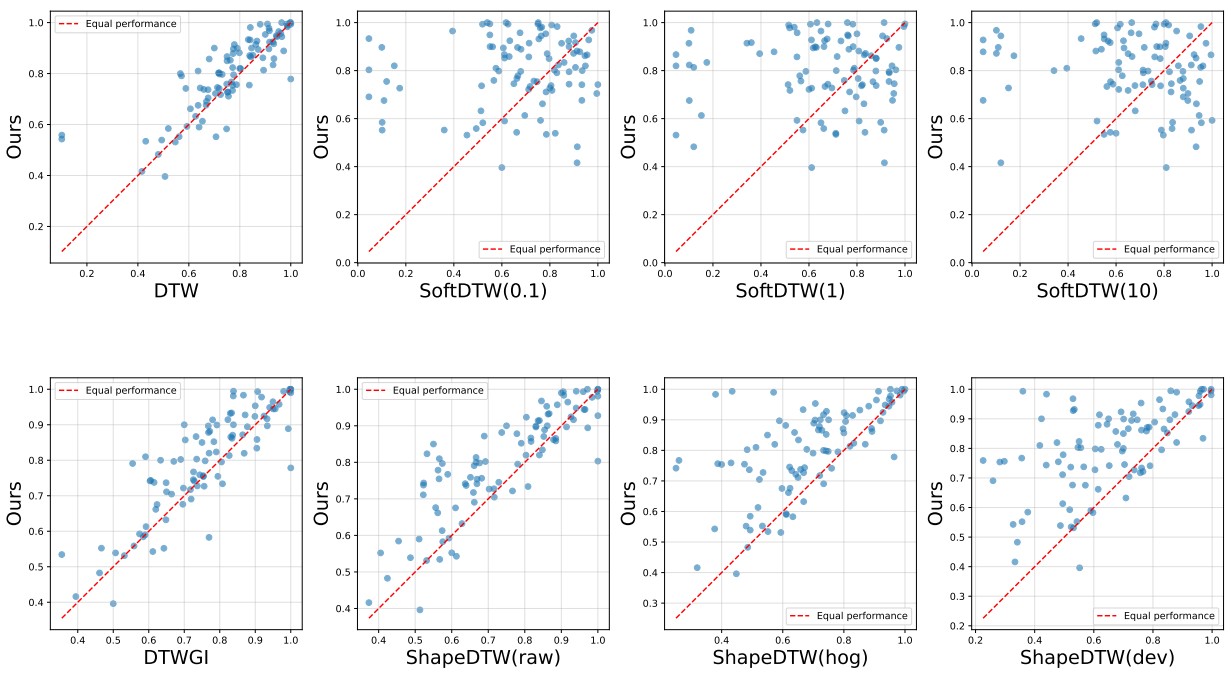

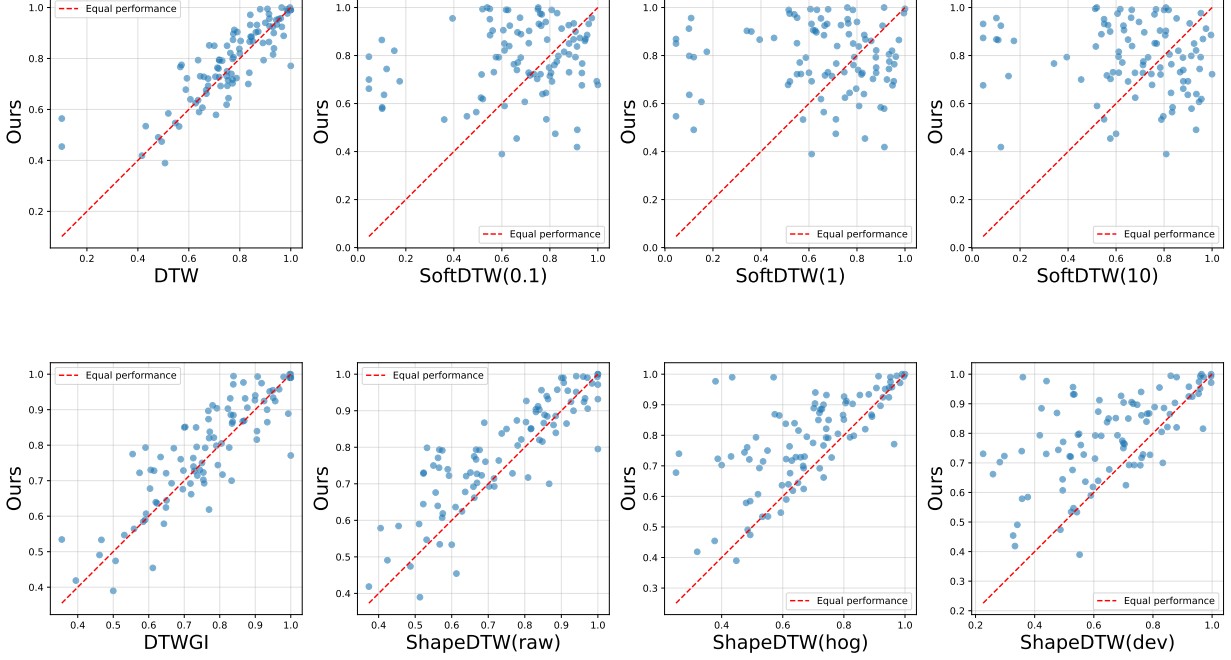

*Figure A.2.* **TP with 100%** of the keypoints vs. several DTW-based methods. Synthetic data training only. Each dot represents a dataset.

*Figure A.3.* **TP with 20%** of the keypoints vs. several DTW-based methods. Synthetic data training only. Each dot represents a dataset.

### A.3. GPU RAM Consumption Analysis

*Table A.1.* **Total GPU memory usage comparison**. Summary of seven UCR datasets and the corresponding GPU memory usage (in GB, rounded to integers) for 1-NN DTW. The *TP + DTW* columns represent the memory footprint at different keypoint ratios, while standard DTW uses the full signal length.

| Dataset | Train | Test | Length | DTW | TP + DTW | | | |
|---|---|---|---|---|---|---|---|---|
| | | | | | 10% | 20% | 50% | 100% |
| ChlorineConcentration | 467 | 3840 | 166 | 186 | 2 | 8 | 47 | 186 |
| Crop | 7200 | 16800 | 46 | 996 | 17 | 46 | 260 | 996 |
| ECG5000 | 500 | 4500 | 140 | 167 | 2 | 7 | 42 | 167 |
| TwoPatterns | 1000 | 4000 | 128 | 248 | 3 | 11 | 63 | 248 |
| UWaveGestureLibraryX | 896 | 3582 | 315 | 1194 | 13 | 49 | 302 | 1194 |
| Wafer | 1000 | 6164 | 152 | 538 | 6 | 22 | 136 | 538 |
| Yoga | 300 | 3000 | 426 | 611 | 7 | 25 | 154 | 611 |

**Memory Consumption and Channel Independence.**  As shown in Table A.1, when both DTW and TP+DTW are applied to the full sequence (keypoint ratio $= 100\%$), the GPU memory consumption remains the same despite the fact that TP features are 256-dimensional. The key reason lies in the memory structure of our DTW implementation.

**Dynamic Programming (DP) Matrix.**  The largest tensor created during DTW is a 4D DP matrix `D` of shape $[\texttt{batch\_size\_x}, \texttt{batch\_size\_y}, \texttt{length\_x} + 1, \texttt{length\_y} + 1]$. Notably, this matrix does *not* include any channel dimension. As a result, increasing the feature dimensionality from, say, 1 to 256 does not change the DP matrix size.

**Cost Computation Over Channels.** At each time step $(i, j)$, we compute a scalar cost by reducing across the channel dimension, whether using cosine similarity or $\ell_2$ distance. Consequently, the DP matrix stores only a single cost value for each pair $(i, j)$, independent of the descriptor dimensionality.

**Equal Memory Footprint at Full Length.** When keypoint ratio $= 100\%$, TP+DTW and plain DTW both operate on the entire time series. Although *TimePoint* descriptors have a higher channel count (e.g., 256), that extra dimensionality is collapsed into a single scalar during the pairwise cost computation, leading to an identical memory footprint for the DP matrix. Hence, the total GPU RAM consumption scales only with the product of the sequence lengths and batch sizes, rather than the descriptor dimensionality or number of channels.

# B. SynthAlign

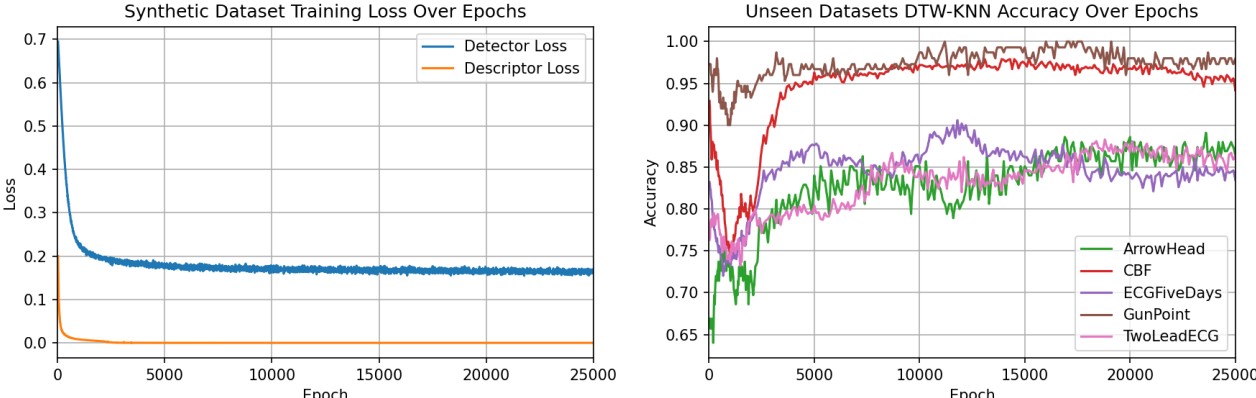

*Figure B.1.* Left: Training losses on `SynthAlign` data. Right: DTW-KNN accuracy on **unseen** datasets. As the training losses decrease the accuracy increases, indicating the loss functions, along with the synthetic training framework (`SynthAlign`) generalize well to real-world data for the task of identifying keypoints and learning descriptors. Epochs are trimmed to 25K for visibility purposes.

**Does synthetic training generalize to real-world data?** A notable finding is that *jointly minimizing the keypoint detection and descriptor losses on the synthetic dataset* leads to improved *k*-NN accuracy on *unseen* real-world data, as evident from Figure B.1. This suggests that learning to detect salient points and produce consistent descriptors under controlled, yet varied, synthetic data and distortions allows for alignment capabilities that transfer beyond the training distribution.

**Data generation** To train and evaluate our method under diverse temporal patterns and keypoint structures, we introduce the `SynthAlign` dataset. Each sample comprises a univariate time series of length $L$ and an associated keypoint (KP) mask that marks salient events (e.g., peaks, start/end points). We generate data on-the-fly using a composition of procedurally defined waveforms and optional augmentations.

`SynthAlign` randomly draws from four principal waveform generators with specified probabilities:

1. *Sine Wave Composition*: Superposes multiple sine waves (random frequency, amplitude, and phase), with derivative-based KPs for local maxima or minima.

2. *Block Wave*: Creates square-wave-like segments with variable block sizes and amplitude, marking boundaries as KPs.

3. *Sawtooth Wave*: Forms a sawtooth signal of random frequency, designating signal resets as KPs.

4. *Radial Basis Function (RBF)*: Summation of Gaussian "blobs," entered at a random position; KPs appear at blob peaks.

Each generated signal has length $L = 512$ by default, though the code supports arbitrary lengths.

**Composition and Augmentations.** After selecting one or more waveform generators (drawn with probabilities $[0.6, 0.15, 0.05, 0.2]$ from the waveforms mentioned in the list above), the resulting signals are summed to form a final sample. We also introduce:

- **Linear Trends:** Randomly superimposed slopes and intercepts to simulate mild non-stationarity.

- **Flips:** Inverts a random subsection of the signal.

- **Noise:** Adds Gaussian noise sampled from $\mathcal{N}(0, 0.1)$ to further diversify training samples.

**Keypoint Extraction.** Keypoints originate from local maxima/minima (*sine* and *sawtooth*), block boundaries, Gaussian centers (*RBF*), and explicit markings for flips and linear boundaries. We ensure start/end points are included only when warranted by the signal design. This strategy provides a rich variety of salient events, allowing models to learn robust keypoint detection across diverse waveform shapes. Overall, SynthAlign delivers a flexible pipeline for generating synthetic time series with automatically labeled keypoints, serving as a valuable testbed for alignment, detection, and descriptor-learning techniques. Figure B.2 shows additional sample drawn from SynthAlign

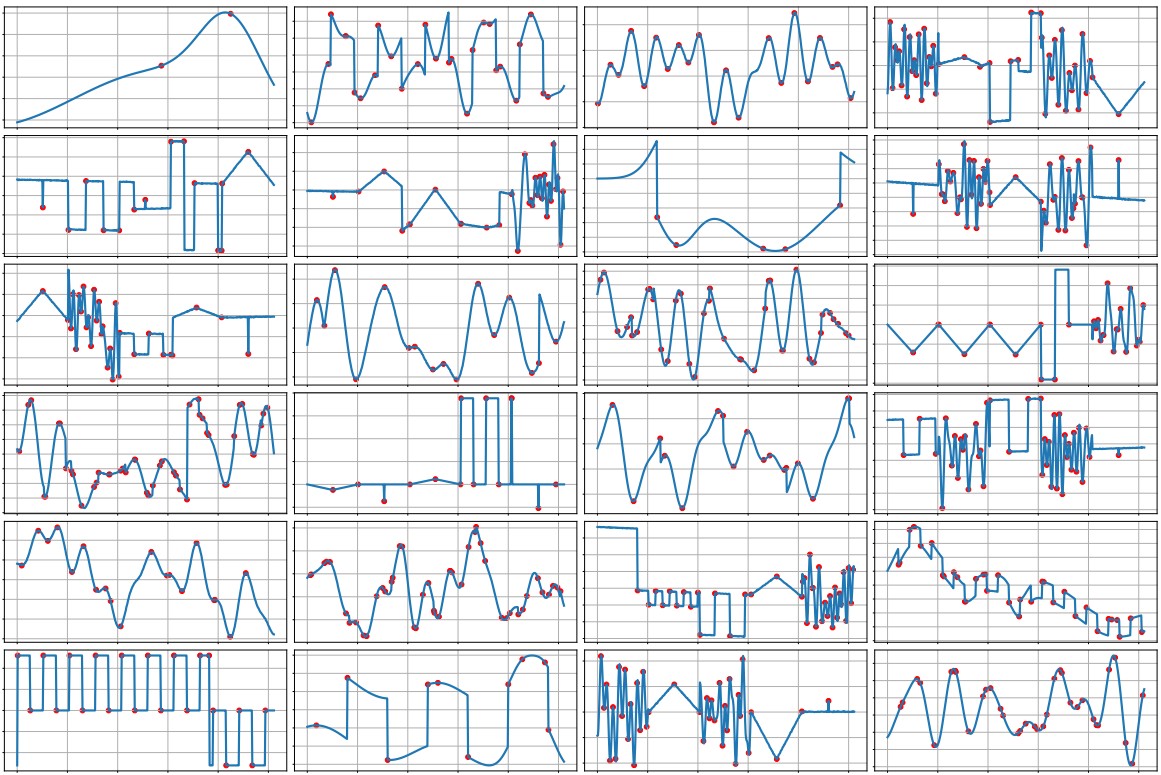

*Figure B.2.* More samples from SynthAlign.

# C. CPAB Transformations

Let $T^{\boldsymbol{\theta}}$ be a diffeomorphism parameterized by $\boldsymbol{\theta}$. The main reason we chose CPAB transformations (Freifeld et al., 2017) as the parametric diffeomorphism family to be used within our method is that they are both expressive and efficient. Our presentation of CPAB transformations below closely follows (Freifeld et al., 2017), though we restrict the discussion to the 1D case (for the more general case, see (Freifeld et al., 2017)).

Let $\Omega = [a, b] \subset \mathbb{R}$ be a finite interval and let $\mathcal{V}$ be a space of continuous functions, from $\Omega$ to $\mathbb{R}$, that are also piecewise-affine w.r.t. some fixed partition of $\Omega$ into sub-intervals. Note that $\mathcal{V}$ is a finite-dimensional linear space. Let $d = \dim(\mathcal{V})$, let $\boldsymbol{\theta} \in \mathbb{R}^d$, and let $v^{\boldsymbol{\theta}} \in \mathcal{V}$ denote the generic element of $\mathcal{V}$, parameterized by $\boldsymbol{\theta}$. The space of CPAB transformations obtained via the integration of elements of $\mathcal{V}$, is defined as

$$\mathcal{T} \triangleq \left\{ T^{\boldsymbol{\theta}} : x \mapsto \phi^{\boldsymbol{\theta}}(x; 1) \text{ s.t. } \phi^{\boldsymbol{\theta}}(x; t) \text{ solves the integral equation } \phi^{\boldsymbol{\theta}}(x; t) = x + \int_0^t v^{\boldsymbol{\theta}}(\phi^{\boldsymbol{\theta}}(x; \tau)) \, \mathrm{d}\tau \text{ where } v^{\boldsymbol{\theta}} \in \mathcal{V} \right\}.$$
(5)

Every $T^{\boldsymbol{\theta}} \in \mathcal{T}$ is an order-preserving transformation (i.e., it is monotonically increasing) and a diffeomorphism (Freifeld et al., 2017). Note that while $v^{\boldsymbol{\theta}} \in \mathcal{V}$ is CPA, the CPAB $T^{\boldsymbol{\theta}} \in \mathcal{T}$ is not (e.g., $T^{\boldsymbol{\theta}}$ is differentiable, unlike any non-trivial CPA function). Equation 5 also implies that the elements of $\mathcal{V}$ are viewed as velocity fields. Particularly useful for us are the following facts: 1) The finer the partition of $\Omega$ is, the more expressive the CPAB family becomes (which also means that $d$ increases). 2) CPAB transformations lend themselves to a fast and accurate computation in closed form of $x \mapsto T^{\boldsymbol{\theta}}(x)$ (Freifeld et al., 2017). Together, these facts mean that *CPAB transformations provide us with a convenient and an efficient way to parameterize nonlinear monotonically-increasing functions.* Figure C.1 show random CPAB transformation applied to synthetic data sampled from SynthAlign, while Figure C.2 shows the effect of increasing the standard deviation ($\sigma_{\mathrm{var}}$) when sampling $\boldsymbol{\theta}$ from the CPA smoothness prior.

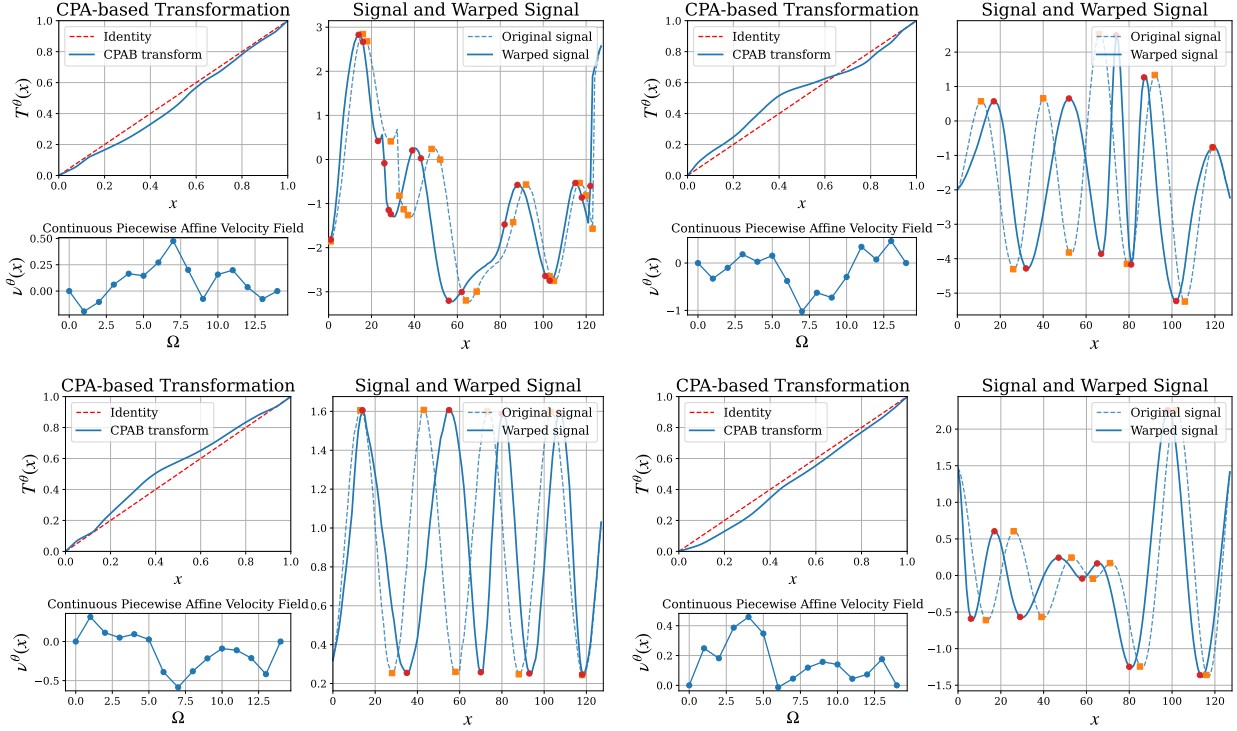

*Figure C.1.* Additional examples of the CPAB transformation.

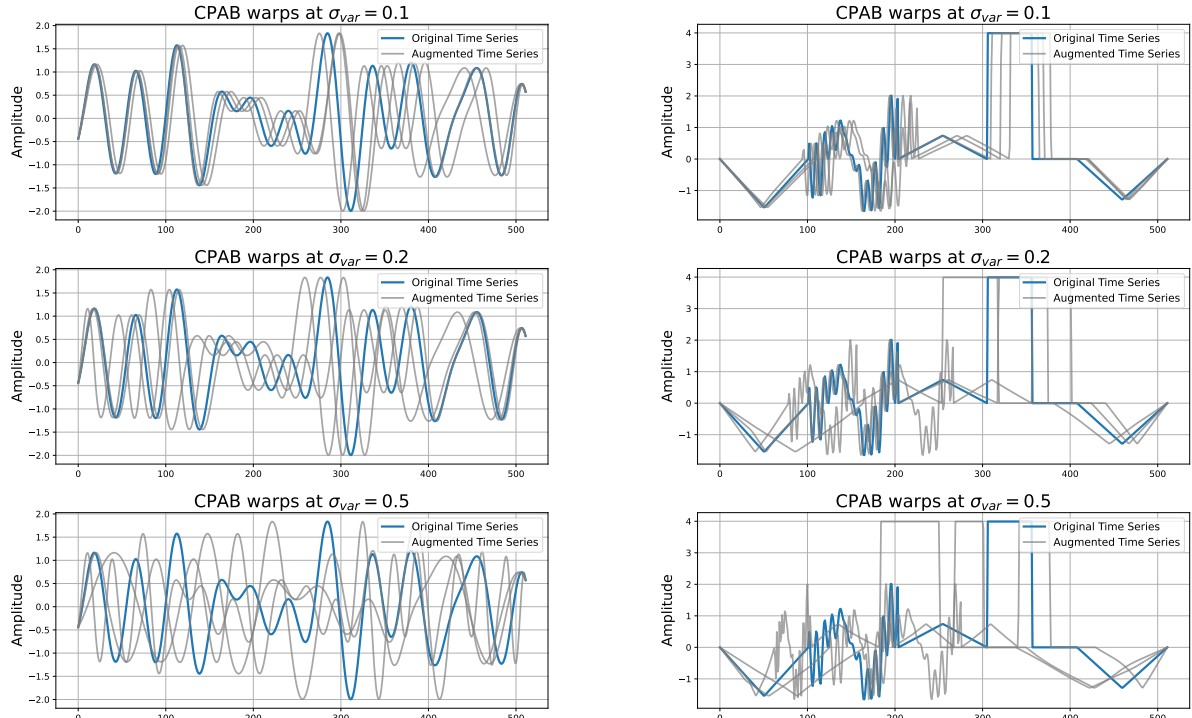

*Figure C.2.* `SynthAlign` synthetic data augmented with CPAB warps at increasing magnitude (top-to-bottom). Each panel shows the original signal in blue and 3 random augmentations in gray.

# D. TimePoint Architecture Details

*Table D.1.* SuperPoint1D model architecture. Conv($C_{in} \rightarrow C_{out}$, $k$) denotes a 1D convolution from $C_{in}$ to $C_{out}$ channels with kernel size $k$, followed by BatchNorm and ReLU. WTConv denotes the wavelet-based convolution with group splitting and scale modules.

| Stage | Layer | Details |
|-------|-------|---------|
| **Encoder** (WTConvEncoder1D) | | |
| *layer1* | ConvBlock1D | • Conv($1 \rightarrow 128$, $k = 3$, stride=1, padding=same)
• BatchNorm(128), ReLU |
| *layer2* | WTConvBlock1D | • WTConv1d($128 \rightarrow 128$, kernel=3, groups=128) + scale modules
• Conv($128 \rightarrow 128$, $k = 1$), BatchNorm(128), ReLU |
| *layer3* | WTConvBlock1D | • WTConv1d($128 \rightarrow 128$, kernel=3, groups=128) + scale modules
• Conv($128 \rightarrow 256$, $k = 1$), BatchNorm(256), ReLU |
| *layer4* | WTConvBlock1D | • WTConv1d($256 \rightarrow 256$, kernel=3, groups=256) + scale modules
• Conv($256 \rightarrow 256$, $k = 1$), BatchNorm(256), ReLU |
| **Detector Head** (DetectorHead1D) | | |
| | Conv($256 \rightarrow 8$, $k = 1$)
Sigmoid | Outputs detector logits for keypoint heatmap.
Normalize the logits to probabilities |
| **Descriptor Head** (DescriptorHead1D) | | |
| | Conv($256 \rightarrow 256$, $k = 1$)
Upsample | Descriptor feature map.
Upsample with scale factor = 8, mode=linear. |

# E. Full UCR Archive Results

## E.1. UCR Archive Details

| ID | Type | Name | Train | Test | Class | Length |
|----|------|------|-------|------|-------|--------|
| 1 | Image | Adiac | 390 | 391 | 37 | 176 |
| 2 | Image | ArrowHead | 36 | 175 | 3 | 251 |
| 3 | Spectro | Beef | 30 | 30 | 5 | 470 |
| 4 | Image | BeetleFly | 20 | 20 | 2 | 512 |
| 5 | Image | BirdChicken | 20 | 20 | 2 | 512 |
| 6 | Sensor | Car | 60 | 60 | 4 | 577 |
| 7 | Simulated | CBF | 30 | 900 | 3 | 128 |
| 8 | Sensor | ChlorineConcentration | 467 | 3840 | 3 | 166 |
| 9 | Sensor | CinCECGTorso | 40 | 1380 | 4 | 1639 |
| 10 | Spectro | Coffee | 28 | 28 | 2 | 286 |
| 11 | Device | Computers | 250 | 250 | 2 | 720 |
| 12 | Motion | CricketX | 390 | 390 | 12 | 300 |
| 13 | Motion | CricketY | 390 | 390 | 12 | 300 |
| 14 | Motion | CricketZ | 390 | 390 | 12 | 300 |
| 15 | Image | DiatomSizeReduction | 16 | 306 | 4 | 345 |
| 16 | Image | DistalPhalanxOutlineAgeGroup | 400 | 139 | 3 | 80 |
| 17 | Image | DistalPhalanxOutlineCorrect | 600 | 276 | 2 | 80 |
| 18 | Image | DistalPhalanxTW | 400 | 139 | 6 | 80 |
| 19 | Sensor | Earthquakes | 322 | 139 | 2 | 512 |
| 20 | ECG | ECG200 | 100 | 100 | 2 | 96 |
| 21 | ECG | ECG5000 | 500 | 4500 | 5 | 140 |
| 22 | ECG | ECGFiveDays | 23 | 861 | 2 | 136 |
| 23 | Device | ElectricDevices | 8926 | 7711 | 7 | 96 |
| 24 | Image | FaceAll | 560 | 1690 | 14 | 131 |
| 25 | Image | FaceFour | 24 | 88 | 4 | 350 |
| 26 | Image | FacesUCR | 200 | 2050 | 14 | 131 |
| 27 | Image | FiftyWords | 450 | 455 | 50 | 270 |
| 28 | Image | Fish | 175 | 175 | 7 | 463 |
| 29 | Sensor | FordA | 3601 | 1320 | 2 | 500 |
| 30 | Sensor | FordB | 3636 | 810 | 2 | 500 |
| 31 | Motion | GunPoint | 50 | 150 | 2 | 150 |
| 32 | Spectro | Ham | 109 | 105 | 2 | 431 |
| 33 | Image | HandOutlines | 1000 | 370 | 2 | 2709 |
| 34 | Motion | Haptics | 155 | 308 | 5 | 1092 |
| 35 | Image | Herring | 64 | 64 | 2 | 512 |
| 36 | Motion | InlineSkate | 100 | 550 | 7 | 1882 |
| 37 | Sensor | InsectWingbeatSound | 220 | 1980 | 11 | 256 |
| 38 | Sensor | ItalyPowerDemand | 67 | 1029 | 2 | 24 |
| 39 | Device | LargeKitchenAppliances | 375 | 375 | 3 | 720 |
| 40 | Sensor | Lightning2 | 60 | 61 | 2 | 637 |
| 41 | Sensor | Lightning7 | 70 | 73 | 7 | 319 |
| 42 | Simulated | Mallat | 55 | 2345 | 8 | 1024 |
| 43 | Spectro | Meat | 60 | 60 | 3 | 448 |
| 44 | Image | MedicalImages | 381 | 760 | 10 | 99 |
| 45 | Image | MiddlePhalanxOutlineAgeGroup | 400 | 154 | 3 | 80 |
| 46 | Image | MiddlePhalanxOutlineCorrect | 600 | 291 | 2 | 80 |
| 47 | Image | MiddlePhalanxTW | 399 | 154 | 6 | 80 |
| 48 | Sensor | MoteStrain | 20 | 1252 | 2 | 84 |
| 49 | ECG | NonInvasiveFetalECGThorax1 | 1800 | 1965 | 42 | 750 |
| 50 | ECG | NonInvasiveFetalECGThorax2 | 1800 | 1965 | 42 | 750 |
| 51 | Spectro | OliveOil | 30 | 30 | 4 | 570 |
| 52 | Image | OSULeaf | 200 | 242 | 6 | 427 |
| 53 | Image | PhalangesOutlinesCorrect | 1800 | 858 | 2 | 80 |
| 54 | Sensor | Phoneme | 214 | 1896 | 39 | 1024 |
| 55 | Sensor | Plane | 105 | 105 | 7 | 144 |
| 56 | Image | ProximalPhalanxOutlineAgeGroup | 400 | 205 | 3 | 80 |
| 57 | Image | ProximalPhalanxOutlineCorrect | 600 | 291 | 2 | 80 |
| 58 | Image | ProximalPhalanxTW | 400 | 205 | 6 | 80 |
| 59 | Device | RefrigerationDevices | 375 | 375 | 3 | 720 |
| 60 | Device | ScreenType | 375 | 375 | 3 | 720 |
| 61 | Simulated | ShapeletSim | 20 | 180 | 2 | 500 |
| 62 | Image | ShapesAll | 600 | 600 | 60 | 512 |
| 63 | Device | SmallKitchenAppliances | 375 | 375 | 3 | 720 |
| 64 | Sensor | SonyAIBORobotSurface1 | 20 | 601 | 2 | 70 |
| 65 | Sensor | SonyAIBORobotSurface2 | 27 | 953 | 2 | 65 |
| 66 | Sensor | StarLightCurves | 1000 | 8236 | 3 | 1024 |
| 67 | Spectro | Strawberry | 613 | 370 | 2 | 235 |
| 68 | Image | SwedishLeaf | 500 | 625 | 15 | 128 |
| 69 | Image | Symbols | 25 | 995 | 6 | 398 |
| 70 | Simulated | SyntheticControl | 300 | 300 | 6 | 60 |
| 71 | Motion | ToeSegmentation1 | 40 | 228 | 2 | 277 |
| 72 | Motion | ToeSegmentation2 | 36 | 130 | 2 | 343 |
| 73 | Sensor | Trace | 100 | 100 | 4 | 275 |
| 74 | ECG | TwoLeadECG | 23 | 1139 | 2 | 82 |

| ID | Type | Name | Train | Test | Class | Length |
|---|---|---|---|---|---|---|
| 75 | Simulated | TwoPatterns | 1000 | 4000 | 4 | 128 |
| 76 | Motion | UWaveGestureLibraryAll | 896 | 3582 | 8 | 945 |
| 77 | Motion | UWaveGestureLibraryX | 896 | 3582 | 8 | 315 |
| 78 | Motion | UWaveGestureLibraryY | 896 | 3582 | 8 | 315 |
| 79 | Motion | UWaveGestureLibraryZ | 896 | 3582 | 8 | 315 |
| 80 | Sensor | Wafer | 1000 | 6164 | 2 | 152 |
| 81 | Spectro | Wine | 57 | 54 | 2 | 234 |
| 82 | Image | WordSynonyms | 267 | 638 | 25 | 270 |
| 83 | Motion | Worms | 181 | 77 | 5 | 900 |
| 84 | Motion | WormsTwoClass | 181 | 77 | 2 | 900 |
| 85 | Image | Yoga | 300 | 3000 | 2 | 426 |
| 86 | Device | ACSF1 | 100 | 100 | 10 | 1460 |
| 87 | Sensor | AllGestureWiimoteX | 300 | 700 | 10 | Vary |
| 88 | Sensor | AllGestureWiimoteY | 300 | 700 | 10 | Vary |
| 89 | Sensor | AllGestureWiimoteZ | 300 | 700 | 10 | Vary |
| 90 | Simulated | BME | 30 | 150 | 3 | 128 |
| 91 | Traffic | Chinatown | 20 | 343 | 2 | 24 |
| 92 | Image | Crop | 7200 | 16800 | 24 | 46 |
| 93 | Sensor | DodgerLoopDay | 78 | 80 | 7 | 288 |
| 94 | Sensor | DodgerLoopGame | 20 | 138 | 2 | 288 |
| 95 | Sensor | DodgerLoopWeekend | 20 | 138 | 2 | 288 |
| 96 | EOG | EOGHorizontalSignal | 362 | 362 | 12 | 1250 |
| 97 | EOG | EOGVerticalSignal | 362 | 362 | 12 | 1250 |
| 98 | Spectro | EthanolLevel | 504 | 500 | 4 | 1751 |
| 99 | Sensor | FreezerRegularTrain | 150 | 2850 | 2 | 301 |
| 100 | Sensor | FreezerSmallTrain | 28 | 2850 | 2 | 301 |
| 101 | HRM | Fungi | 18 | 186 | 18 | 201 |
| 102 | Trajectory | GestureMidAirD1 | 208 | 130 | 26 | Vary |
| 103 | Trajectory | GestureMidAirD2 | 208 | 130 | 26 | Vary |
| 104 | Trajectory | GestureMidAirD3 | 208 | 130 | 26 | Vary |
| 105 | Sensor | GesturePebbleZ1 | 132 | 172 | 6 | Vary |
| 106 | Sensor | GesturePebbleZ2 | 146 | 158 | 6 | Vary |
| 107 | Motion | GunPointAgeSpan | 135 | 316 | 2 | 150 |
| 108 | Motion | GunPointMaleVersusFemale | 135 | 316 | 2 | 150 |
| 109 | Motion | GunPointOldVersusYoung | 136 | 315 | 2 | 150 |
| 110 | Device | HouseTwenty | 40 | 119 | 2 | 2000 |
| 111 | EPG | InsectEPGRegularTrain | 62 | 249 | 3 | 601 |
| 112 | EPG | InsectEPGSmallTrain | 17 | 249 | 3 | 601 |
| 113 | Traffic | MelbournePedestrian | 1194 | 2439 | 10 | 24 |
| 114 | Image | MixedShapesRegularTrain | 500 | 2425 | 5 | 1024 |
| 115 | Image | MixedShapesSmallTrain | 100 | 2425 | 5 | 1024 |
| 116 | Sensor | PickupGestureWiimoteZ | 50 | 50 | 10 | Vary |
| 117 | Hemodynamics | PigAirwayPressure | 104 | 208 | 52 | 2000 |
| 118 | Hemodynamics | PigArtPressure | 104 | 208 | 52 | 2000 |
| 119 | Hemodynamics | PigCVP | 104 | 208 | 52 | 2000 |
| 120 | Device | PLAID | 537 | 537 | 11 | Vary |
| 121 | Power | PowerCons | 180 | 180 | 2 | 144 |
| 122 | Spectrum | Rock | 20 | 50 | 4 | 2844 |
| 123 | Spectrum | SemgHandGenderCh2 | 300 | 600 | 2 | 1500 |
| 124 | Spectrum | SemgHandMovementCh2 | 450 | 450 | 6 | 1500 |
| 125 | Spectrum | SemgHandSubjectCh2 | 450 | 450 | 5 | 1500 |
| 126 | Sensor | ShakeGestureWiimoteZ | 50 | 50 | 10 | Vary |
| 127 | Simulated | SmoothSubspace | 150 | 150 | 3 | 15 |
| 128 | Simulated | UMD | 36 | 144 | 3 | 150 |

## E.2. UCR Archive Details

Table E.2: Accuracy Comparison Between TimePoint Configurations (`SynthAlign` training only.)

| Dataset/Method | TP+DTW | | | | TP+SoftDTW ($\gamma = 1$) | | | |
|---|---|---|---|---|---|---|---|---|
| | (0.1) | (0.2) | (0.5) | (1) | (0.1) | (0.2) | (0.5) | (1) |
| Adiac | 0.645 | 0.678 | 0.734 | 0.742 | 0.519 | 0.445 | 0.598 | 0.737 |
| AllGestureWiimoteX | 0.55 | 0.454 | 0.552 | 0.542 | 0.461 | 0.421 | 0.454 | 0.437 |
| ArrowHead | 0.869 | 0.851 | 0.874 | 0.857 | 0.669 | 0.657 | 0.806 | 0.840 |
| BME | 0.960 | 0.940 | 0.893 | 0.953 | 0.953 | 0.893 | 0.860 | 0.947 |
| Beef | 0.733 | 0.767 | 0.700 | 0.800 | 0.600 | 0.500 | 0.700 | 0.833 |
| BeetleFly | 0.950 | 0.850 | 0.900 | 0.900 | 0.850 | 0.950 | 0.850 | 0.950 |
| BirdChicken | 0.900 | 0.750 | 0.800 | 0.850 | 0.900 | 0.750 | 0.950 | 0.750 |
| CBF | 0.940 | 0.990 | 0.940 | 0.993 | 0.930 | 0.970 | 0.988 | 0.996 |
| Car | 0.850 | 0.850 | 0.767 | 0.867 | 0.633 | 0.733 | 0.783 | 0.817 |
| Chinatown | 0.831 | 0.924 | 0.915 | 0.945 | 0.828 | 0.901 | 0.974 | 0.942 |
| ChlorineConcentration | 0.574 | 0.624 | 0.648 | 0.632 | 0.527 | 0.538 | 0.574 | 0.635 |
| Coffee | 1.000 | 1.000 | 1.000 | 1.000 | 0.750 | 0.714 | 0.893 | 1.000 |
| Computers | 0.668 | 0.676 | 0.664 | 0.676 | 0.628 | 0.664 | 0.620 | 0.552 |
| CricketX | 0.715 | 0.703 | 0.551 | 0.754 | 0.610 | 0.585 | 0.685 | 0.685 |
| CricketY | 0.664 | 0.731 | 0.582 | 0.759 | 0.590 | 0.628 | 0.705 | 0.715 |
| CricketZ | 0.726 | 0.723 | 0.590 | 0.756 | 0.654 | 0.592 | 0.713 | 0.708 |
| Crop | 0.665 | 0.693 | 0.682 | 0.705 | 0.658 | 0.670 | 0.693 | 0.705 |
| DiatomSizeReduction | 0.967 | 0.958 | 0.967 | 0.958 | 0.850 | 0.794 | 0.905 | 0.958 |
| DistalPhalanxOutlineAgeGroup | 0.583 | 0.619 | 0.633 | 0.583 | 0.583 | 0.604 | 0.612 | 0.547 |
| DistalPhalanxOutlineCorrect | 0.652 | 0.692 | 0.699 | 0.717 | 0.616 | 0.685 | 0.678 | 0.714 |
| DistalPhalanxTW | 0.511 | 0.590 | 0.583 | 0.590 | 0.554 | 0.532 | 0.540 | 0.576 |
| ECG200 | 0.850 | 0.820 | 0.820 | 0.820 | 0.820 | 0.820 | 0.900 | 0.800 |
| ECG5000 | 0.921 | 0.924 | 0.921 | 0.924 | 0.922 | 0.916 | 0.922 | 0.924 |
| ECGFiveDays | 0.812 | 0.897 | 0.942 | 0.897 | 0.659 | 0.683 | 0.676 | 0.895 |
| Earthquakes | 0.669 | 0.662 | 0.691 | 0.691 | 0.676 | 0.662 | 0.619 | 0.676 |
| ElectricDevices | 0.593 | 0.607 | 0.616 | 0.613 | 0.588 | 0.592 | 0.602 | 0.615 |
| FaceAll | 0.676 | 0.717 | 0.720 | 0.734 | 0.634 | 0.667 | 0.691 | 0.730 |
| FaceFour | 0.807 | 0.932 | 0.886 | 0.932 | 0.693 | 0.727 | 0.932 | 0.943 |
| FacesUCR | 0.735 | 0.840 | 0.802 | 0.859 | 0.682 | 0.762 | 0.831 | 0.852 |
| FiftyWords | 0.745 | 0.760 | 0.736 | 0.802 | 0.684 | 0.668 | 0.677 | 0.754 |
| Fish | 0.874 | 0.903 | 0.880 | 0.914 | 0.731 | 0.709 | 0.834 | 0.891 |
| FordA | 0.780 | 0.775 | 0.768 | 0.791 | 0.717 | 0.729 | 0.754 | 0.782 |
| FordB | 0.641 | 0.640 | 0.647 | 0.662 | 0.586 | 0.633 | 0.640 | 0.623 |
| FreezerRegularTrain | 0.913 | 0.927 | 0.876 | 0.924 | 0.891 | 0.876 | 0.981 | 0.920 |
| FreezerSmallTrain | 0.780 | 0.793 | 0.760 | 0.798 | 0.758 | 0.783 | 0.862 | 0.771 |
| Fungi | 0.957 | 0.995 | 0.925 | 0.995 | 0.839 | 0.731 | 0.957 | 0.984 |
| GestureMidAirD1 | 0.038 | 0.038 | 0.038 | 0.038 | 0.038 | 0.038 | 0.038 | 0.038 |
| GestureMidAirD2 | 0.038 | 0.038 | 0.038 | 0.038 | 0.038 | 0.038 | 0.038 | 0.038 |
| GestureMidAirD3 | 0.038 | 0.038 | 0.038 | 0.038 | 0.038 | 0.038 | 0.038 | 0.038 |
| GesturePebbleZ1 | 0.163 | 0.163 | 0.163 | 0.163 | 0.163 | 0.163 | 0.163 | 0.163 |
| GesturePebbleZ2 | 0.152 | 0.152 | 0.152 | 0.152 | 0.152 | 0.152 | 0.152 | 0.152 |
| GunPoint | 0.973 | 0.993 | 0.973 | 0.993 | 0.913 | 0.860 | 0.980 | 0.993 |
| GunPointAgeSpan | 0.981 | 0.975 | 0.972 | 0.978 | 0.953 | 0.924 | 0.972 | 0.965 |
| GunPointMaleVersusFemale | 0.994 | 1.000 | 0.997 | 1.000 | 0.978 | 0.994 | 1.000 | 1.000 |
| GunPointOldVersusYoung | 0.968 | 0.971 | 0.959 | 0.981 | 0.895 | 0.940 | 0.971 | 0.971 |
| Ham | 0.514 | 0.533 | 0.457 | 0.552 | 0.562 | 0.524 | 0.600 | 0.533 |
| Herring | 0.469 | 0.547 | 0.531 | 0.531 | 0.594 | 0.625 | 0.453 | 0.516 |
| InsectEPGRegularTrain | 0.964 | 0.932 | 0.876 | 0.928 | 0.855 | 0.827 | 0.807 | 0.819 |
| AllGestureWiimoteY | 0.558 | 0.564 | 0.578 | 0.558 | 0.492 | 0.47 | 0.47 | 0.457 |
| AllGestureWiimoteZ | 0.282 | 0.237 | 0.287 | 0.28 | 0.212 | 0.227 | 0.177 | 0.172 |
| InsectEPGSmallTrain | 0.815 | 0.795 | 0.703 | 0.803 | 0.622 | 0.755 | 0.711 | 0.747 |
| InsectWingbeatSound | 0.468 | 0.534 | 0.504 | 0.534 | 0.427 | 0.487 | 0.554 | 0.563 |
| ItalyPowerDemand | 0.941 | 0.934 | 0.957 | 0.945 | 0.943 | 0.939 | 0.944 | 0.949 |
| LargeKitchenAppliances | 0.757 | 0.744 | 0.709 | 0.755 | 0.640 | 0.613 | 0.589 | 0.560 |
| Lightning2 | 0.836 | 0.869 | 0.852 | 0.820 | 0.738 | 0.803 | 0.820 | 0.852 |
| Lightning7 | 0.671 | 0.740 | 0.808 | 0.767 | 0.616 | 0.658 | 0.658 | 0.740 |
| Mallat | 0.912 | 0.865 | 0.807 | 0.897 | 0.664 | 0.630 | 0.782 | 0.900 |
| Meat | 0.900 | 0.900 | 0.883 | 0.917 | 0.783 | 0.800 | 0.800 | 0.883 |
| MedicalImages | 0.687 | 0.714 | 0.729 | 0.728 | 0.674 | 0.672 | 0.722 | 0.714 |
| MiddlePhalanxOutlineAgeGroup | 0.422 | 0.390 | 0.461 | 0.396 | 0.364 | 0.429 | 0.396 | 0.442 |
| MiddlePhalanxOutlineCorrect | 0.663 | 0.729 | 0.694 | 0.722 | 0.625 | 0.601 | 0.622 | 0.722 |
| MiddlePhalanxTW | 0.442 | 0.474 | 0.539 | 0.539 | 0.422 | 0.487 | 0.526 | 0.578 |
| MixedShapesSmallTrain | 0.884 | 0.912 | 0.889 | 0.914 | 0.798 | 0.874 | 0.831 | 0.836 |
| MoteStrain | 0.826 | 0.861 | 0.852 | 0.862 | 0.807 | 0.863 | 0.865 | 0.856 |
| NonInvasiveFetalECGThorax1 | 0.814 | 0.798 | 0.769 | 0.823 | 0.583 | 0.593 | 0.751 | 0.814 |
| NonInvasiveFetalECGThorax2 | 0.847 | 0.867 | 0.846 | 0.872 | 0.693 | 0.677 | 0.825 | 0.860 |
| OSULeaf | 0.789 | 0.793 | 0.769 | 0.810 | 0.657 | 0.686 | 0.661 | 0.715 |
| OliveOil | 0.567 | 0.700 | 0.900 | 0.933 | 0.633 | 0.400 | 0.600 | 0.633 |
| PhalangesOutlinesCorrect | 0.690 | 0.727 | 0.748 | 0.741 | 0.671 | 0.697 | 0.681 | 0.754 |
| Phoneme | 0.267 | 0.268 | 0.261 | 0.284 | 0.212 | 0.203 | 0.166 | 0.165 |
| PickupGestureWiimoteZ | 0.100 | 0.100 | 0.100 | 0.100 | 0.100 | 0.100 | 0.100 | 0.100 |
| Plane | 1.000 | 0.990 | 1.000 | 1.000 | 1.000 | 0.981 | 1.000 | 0.990 |
| PowerCons | 0.856 | 0.906 | 0.878 | 0.894 | 0.811 | 0.867 | 0.911 | 0.861 |
| ProximalPhalanxOutlineAgeGroup | 0.790 | 0.805 | 0.805 | 0.795 | 0.810 | 0.780 | 0.800 | 0.800 |
| ProximalPhalanxOutlineCorrect | 0.818 | 0.821 | 0.825 | 0.852 | 0.832 | 0.821 | 0.818 | 0.845 |
| ProximalPhalanxTW | 0.673 | 0.693 | 0.712 | 0.727 | 0.683 | 0.727 | 0.751 | 0.712 |
| RefrigerationDevices | 0.525 | 0.491 | 0.504 | 0.483 | 0.461 | 0.451 | 0.429 | 0.483 |
| ScreenType | 0.413 | 0.419 | 0.411 | 0.416 | 0.376 | 0.397 | 0.387 | 0.384 |
| ShakeGestureWiimoteZ | 0.100 | 0.100 | 0.100 | 0.100 | 0.100 | 0.100 | 0.100 | 0.100 |
| ShapeletSim | 0.600 | 0.644 | 0.617 | 0.711 | 0.533 | 0.578 | 0.550 | 0.644 |
| ShapesAll | 0.860 | 0.873 | 0.862 | 0.878 | 0.728 | 0.782 | 0.777 | 0.803 |
| SmallKitchenAppliances | 0.555 | 0.579 | 0.600 | 0.552 | 0.557 | 0.576 | 0.507 | 0.475 |
| SmoothSubspace | 0.800 | 0.793 | 0.807 | 0.813 | 0.793 | 0.807 | 0.807 | 0.793 |
| SonyAIBORobotSurface1 | 0.839 | 0.764 | 0.760 | 0.745 | 0.844 | 0.800 | 0.785 | 0.737 |
| SonyAIBORobotSurface2 | 0.860 | 0.886 | 0.890 | 0.866 | 0.848 | 0.871 | 0.855 | 0.860 |
| Strawberry | 0.932 | 0.951 | 0.943 | 0.949 | 0.870 | 0.889 | 0.927 | 0.951 |

| Dataset/Method | TP+DTW | | | | TP+SoftDTW ($\gamma = 1$) | | | |
|---|---|---|---|---|---|---|---|---|
| | (0.1) | (0.2) | (0.5) | (1) | (0.1) | (0.2) | (0.5) | (1) |
| SwedishLeaf | 0.875 | 0.904 | 0.888 | 0.899 | 0.790 | 0.811 | 0.866 | 0.904 |
| Symbols | 0.949 | 0.955 | 0.939 | 0.965 | 0.897 | 0.912 | 0.860 | 0.930 |
| SyntheticControl | 0.957 | 0.977 | 0.967 | 0.983 | 0.967 | 0.967 | 0.970 | 0.980 |
| ToeSegmentation1 | 0.846 | 0.838 | 0.855 | 0.882 | 0.789 | 0.785 | 0.789 | 0.829 |
| ToeSegmentation2 | 0.823 | 0.885 | 0.831 | 0.900 | 0.777 | 0.800 | 0.862 | 0.869 |
| Trace | 0.990 | 0.990 | 0.990 | 0.990 | 1.000 | 0.980 | 0.920 | 0.990 |
| TwoLeadECG | 0.830 | 0.816 | 0.801 | 0.834 | 0.789 | 0.695 | 0.785 | 0.837 |
| TwoPatterns | 0.729 | 0.771 | 0.687 | 0.778 | 0.681 | 0.676 | 0.602 | 0.756 |
| UMD | 0.910 | 0.889 | 0.840 | 0.889 | 0.854 | 0.819 | 0.812 | 0.854 |
| UWaveGestureLibraryAll | 0.917 | 0.956 | 0.927 | 0.968 | 0.784 | 0.913 | 0.910 | 0.905 |
| UWaveGestureLibraryX | 0.758 | 0.791 | 0.786 | 0.797 | 0.715 | 0.753 | 0.740 | 0.788 |
| UWaveGestureLibraryY | 0.703 | 0.730 | 0.716 | 0.743 | 0.643 | 0.682 | 0.660 | 0.711 |
| UWaveGestureLibraryZ | 0.689 | 0.728 | 0.701 | 0.738 | 0.652 | 0.693 | 0.688 | 0.714 |
| Wafer | 0.994 | 0.993 | 0.989 | 0.994 | 0.990 | 0.990 | 0.995 | 0.994 |
| Wine | 0.611 | 0.722 | 0.741 | 0.593 | 0.481 | 0.611 | 0.574 | 0.630 |
| WordSynonyms | 0.688 | 0.721 | 0.694 | 0.737 | 0.594 | 0.605 | 0.589 | 0.694 |
| Worms | 0.662 | 0.584 | 0.610 | 0.584 | 0.649 | 0.519 | 0.558 | 0.597 |
| WormsTwoClass | 0.727 | 0.636 | 0.649 | 0.675 | 0.688 | 0.649 | 0.623 | 0.688 |
| Yoga | 0.858 | 0.866 | 0.853 | 0.871 | 0.768 | 0.760 | 0.832 | 0.856 |

Table E.3: Accuracy Comparison Between Competitors.

| Dataset/Method | DTW | DTW-GI | Euc. | ShapeDTW | | | SoftDTW | | |
|---|---|---|---|---|---|---|---|---|---|
| | | | | (dev) | (hog) | (raw) | ($\gamma = 0.1$) | ($\gamma = 1$) | ($\gamma = 10$) |
| Adiac | 0.588 | 0.604 | 0.066 | 0.652 | 0.251 | 0.637 | 1.000 | 0.513 | 0.750 |
| AllGestureWiimoteX | 0.135 | 0.611 | 0.101 | 0.327 | 0.377 | 0.613 | 0.662 | 0.833 | 0.576 |
| ArrowHead | 0.680 | 0.703 | 0.229 | 0.674 | 0.800 | 0.817 | 0.712 | 0.550 | 0.761 |
| BME | 0.900 | 0.900 | 0.493 | 0.760 | 0.707 | 0.860 | 0.783 | 0.800 | 0.696 |
| Beef | 0.567 | 0.633 | 0.267 | 0.700 | 0.733 | 0.667 | 0.935 | 0.714 | 0.341 |
| BeetleFly | 0.700 | 0.700 | 0.450 | 0.650 | 0.750 | 0.750 | 0.880 | 0.784 | 0.769 |
| BirdChicken | 0.750 | 0.750 | 0.800 | 0.700 | 0.550 | 0.550 | 0.797 | 0.880 | 0.784 |
| CBF | 1.000 | 0.997 | 0.658 | 0.360 | 0.434 | 0.906 | 0.626 | 0.783 | 0.800 |
| Car | 0.750 | 0.733 | 0.600 | 0.717 | 0.717 | 0.817 | 0.610 | 0.046 | 0.925 |
| Chinatown | 0.965 | 0.956 | 0.805 | 0.933 | 0.948 | 0.962 | 0.764 | 0.907 | 0.120 |
| ChlorineConcentration | 0.627 | 0.648 | 0.231 | 0.709 | 0.668 | 0.628 | 0.513 | 0.750 | 0.680 |
| Coffee | 0.964 | 1.000 | 0.536 | 0.964 | 1.000 | 1.000 | 0.752 | 0.913 | 0.808 |
| Computers | 0.668 | 0.696 | 0.672 | 0.528 | 0.624 | 0.556 | 0.933 | 0.948 | 0.046 |
| CricketX | 0.772 | 0.754 | 0.097 | 0.282 | 0.400 | 0.669 | 0.747 | 0.867 | 0.577 |
| CricketY | 0.749 | 0.744 | 0.074 | 0.226 | 0.431 | 0.651 | 0.567 | 0.747 | 0.867 |
| CricketZ | 0.787 | 0.754 | 0.092 | 0.297 | 0.387 | 0.682 | 0.733 | 0.567 | 0.747 |
| Crop | 0.676 | 0.664 | 0.052 | 0.719 | 0.524 | 0.716 | 0.995 | 0.953 | 0.899 |
| DiatomSizeReduction | 0.961 | 0.967 | 0.431 | 0.922 | 0.958 | 0.889 | 0.676 | 0.551 | 0.663 |
| DistalPhalanxOutlineAgeGroup | 0.748 | 0.770 | 0.604 | 0.597 | 0.633 | 0.576 | 0.521 | 0.650 | 0.955 |
| DistalPhalanxOutlineCorrect | 0.725 | 0.717 | 0.478 | 0.757 | 0.721 | 0.659 | 0.809 | 0.521 | 0.650 |
| DistalPhalanxTW | 0.640 | 0.590 | 0.468 | 0.590 | 0.612 | 0.511 | 0.611 | 0.809 | 0.521 |
| ECG200 | 0.800 | 0.770 | 0.350 | 0.880 | 0.870 | 0.840 | 0.152 | 0.946 | 0.859 |
| ECG5000 | 0.930 | 0.925 | 0.137 | 0.921 | 0.928 | 0.926 | 0.633 | 0.665 | 0.631 |
| ECGFiveDays | 0.775 | 0.768 | 0.551 | 0.747 | 0.920 | 0.830 | 0.665 | 0.631 | 0.717 |
| Earthquakes | 0.669 | 0.719 | 0.698 | 0.259 | 0.734 | 0.662 | 0.046 | 0.797 | 0.880 |
| ElectricDevices | 0.653 | 0.592 | 0.232 | 0.495 | 0.519 | 0.574 | 0.696 | 0.152 | 0.946 |
| FaceAll | 0.772 | 0.808 | 0.019 | 0.762 | 0.630 | 0.809 | 0.717 | 0.667 | 0.562 |
| FaceFour | 0.841 | 0.830 | 0.364 | 0.534 | 0.852 | 0.864 | 0.754 | 0.676 | 0.551 |
| FacesUCR | 0.934 | 0.905 | 0.143 | 0.778 | 0.639 | 0.885 | 0.631 | 0.717 | 0.667 |
| FiftyWords | 0.716 | 0.690 | 0.022 | 0.556 | 0.484 | 0.692 | 0.819 | 0.785 | 0.712 |
| Fish | 0.863 | 0.823 | 0.257 | 0.874 | 0.829 | 0.840 | 0.714 | 0.341 | 0.808 |
| FordA | 0.571 | 0.555 | 0.484 | 0.696 | 0.699 | 0.661 | 0.829 | 0.935 | 0.714 |
| FordB | 0.606 | 0.620 | 0.505 | 0.615 | 0.619 | 0.562 | 0.769 | 0.829 | 0.935 |
| FreezerRegularTrain | 0.917 | 0.899 | 0.546 | 0.692 | 0.801 | 0.804 | 0.879 | 0.733 | 0.567 |
| FreezerSmallTrain | 0.720 | 0.759 | 0.703 | 0.605 | 0.742 | 0.676 | 0.587 | 0.879 | 0.733 |
| Fungi | 0.909 | 0.839 | 0.038 | 0.860 | 0.973 | 0.941 | 0.550 | 1.000 | 0.513 |
| GestureMidAirD1 | 0.046 | 0.538 | 0.046 | 0.485 | 0.400 | 0.508 | 0.684 | 0.754 | 0.676 |
| GestureMidAirD2 | 0.046 | 0.438 | 0.046 | 0.338 | 0.431 | 0.454 | 0.739 | 0.684 | 0.754 |
| GestureMidAirD3 | 0.046 | 0.169 | 0.046 | 0.346 | 0.300 | 0.292 | 0.576 | 0.739 | 0.684 |
| GesturePebbleZ1 | 0.174 | 0.616 | 0.174 | 0.174 | 0.581 | 0.750 | 0.808 | 0.907 | 0.360 |
| GesturePebbleZ2 | 0.152 | 0.563 | 0.152 | 0.203 | 0.563 | 0.722 | 0.341 | 0.808 | 0.907 |
| GunPoint | 0.880 | 0.907 | 0.513 | 0.960 | 0.913 | 0.960 | 0.519 | 0.606 | 0.618 |
| GunPointAgeSpan | 0.915 | 0.918 | 0.690 | 0.956 | 0.953 | 0.984 | 0.760 | 0.519 | 0.606 |
| GunPointMaleVersusFemale | 0.997 | 0.997 | 0.867 | 0.997 | 0.991 | 1.000 | 0.539 | 0.760 | 0.519 |
| GunPointOldVersusYoung | 0.841 | 0.838 | 0.514 | 0.997 | 0.984 | 1.000 | 0.766 | 0.539 | 0.760 |
| Ham | 0.562 | 0.467 | 0.486 | 0.543 | 0.533 | 0.600 | 0.360 | 0.575 | 0.789 |
| Herring | 0.547 | 0.531 | 0.406 | 0.531 | 0.594 | 0.803 | 0.455 | 0.046 | 0.797 |
| InsectEPGRegularTrain | 0.867 | 0.871 | 0.703 | 0.530 | 0.743 | 1.000 | 0.962 | 0.610 | 0.046 |
| AllGestureWiimoteY | 0.154 | 0.558 | 0.1 | Nan | Nan | Nan | 0.493 | 0.661 | 0.833 |
| AllGestureWiimoteZ | 0.094 | 0.288 | 0.11 | Nan | Nan | Nan | 0.493 | 0.661 | 0.852 |
| InsectEPGSmallTrain | 0.719 | 0.735 | 0.691 | 0.546 | 0.679 | 1.000 | 0.046 | 0.962 | 0.610 |
| InsectWingbeatSound | 0.431 | 0.355 | 0.091 | 0.523 | 0.552 | 0.567 | 0.785 | 0.712 | 0.550 |
| ItalyPowerDemand | 0.946 | 0.950 | 0.532 | 0.954 | 0.881 | 0.965 | 0.899 | 0.764 | 0.907 |
| LargeKitchenAppliances | 0.837 | 0.795 | 0.355 | 0.480 | 0.475 | 0.565 | 0.120 | 0.933 | 0.948 |
| Lightning2 | 0.803 | 0.869 | 0.541 | 0.475 | 0.574 | 0.803 | 0.948 | 0.046 | 0.962 |
| Lightning7 | 0.767 | 0.726 | 0.151 | 0.356 | 0.260 | 0.589 | 0.663 | 0.805 | 0.880 |
| Mallat | 0.914 | 0.934 | 0.244 | 0.857 | 0.589 | 0.914 | 0.100 | 0.975 | 0.109 |
| Meat | 0.933 | 0.933 | 0.333 | 0.733 | 0.733 | 0.933 | 0.907 | 0.360 | 0.575 |
| MedicalImages | 0.754 | 0.737 | 0.478 | 0.604 | 0.536 | 0.716 | 0.800 | 0.696 | 0.152 |
| MiddlePhalanxOutlineAgeGroup | 0.506 | 0.500 | 0.208 | 0.552 | 0.448 | 0.513 | 0.600 | 0.611 | 0.809 |

**TimePoint**

| Dataset/Method | DTW | DTW-GI | Euc. | ShapeDTW | | | SoftDTW | | |
|---|---|---|---|---|---|---|---|---|---|
| | | | | (dev) | (hog) | (raw) | ($\gamma = 0.1$) | ($\gamma = 1$) | ($\gamma = 10$) |
| MiddlePhalanxOutlineCorrect | 0.704 | 0.698 | 0.584 | 0.766 | 0.643 | 0.766 | 0.712 | 0.600 | 0.611 |
| MiddlePhalanxTW | 0.494 | 0.506 | 0.448 | 0.487 | 0.494 | 0.487 | 0.823 | 0.712 | 0.600 |
| MixedShapesSmallTrain | 0.779 | 0.780 | 0.223 | 0.619 | 0.808 | 0.836 | 0.849 | 0.100 | 0.975 |
| MoteStrain | 0.891 | 0.835 | 0.570 | 0.761 | 0.892 | 0.879 | 0.946 | 0.859 | 0.174 |
| NonInvasiveFetalECGThorax1 | 0.772 | 0.790 | 0.024 | 0.550 | 0.833 | 0.532 | 0.835 | 0.101 | 0.913 |
| NonInvasiveFetalECGThorax2 | 0.851 | 0.864 | 0.032 | 0.787 | 0.891 | 0.689 | 0.899 | 0.835 | 0.101 |
| OSULeaf | 0.636 | 0.591 | 0.198 | 0.417 | 0.512 | 0.566 | 0.575 | 0.789 | 0.395 |
| OliveOil | 0.833 | 0.833 | 0.133 | 0.833 | 0.667 | 0.867 | 0.046 | 0.925 | 0.455 |
| PhalangesOutlinesCorrect | 0.719 | 0.728 | 0.503 | 0.790 | 0.760 | 0.669 | 0.933 | 0.823 | 0.712 |
| Phoneme | 0.272 | 0.228 | 0.011 | 0.082 | 0.047 | 0.117 | 0.952 | 0.849 | 0.100 |
| PickupGestureWiimoteZ | 0.120 | 0.220 | 0.120 | 0.280 | 0.480 | 0.700 | 0.833 | 0.576 | 0.739 |
| Plane | 1.000 | 1.000 | 0.095 | 0.971 | 0.952 | 0.971 | 0.618 | 0.633 | 0.665 |
| PowerCons | 0.872 | 0.878 | 0.506 | 0.728 | 0.806 | 0.972 | 0.606 | 0.618 | 0.633 |
| ProximalPhalanxOutlineAgeGroup | 0.776 | 0.805 | 0.820 | 0.829 | 0.780 | 0.780 | 0.880 | 0.933 | 0.823 |
| ProximalPhalanxOutlineCorrect | 0.763 | 0.784 | 0.550 | 0.849 | 0.742 | 0.790 | 0.611 | 0.880 | 0.933 |
| ProximalPhalanxTW | 0.751 | 0.756 | 0.737 | 0.737 | 0.668 | 0.702 | 0.174 | 0.611 | 0.880 |
| RefrigerationDevices | 0.480 | 0.461 | 0.384 | 0.341 | 0.485 | 0.424 | 0.914 | 0.120 | 0.933 |
| ScreenType | 0.416 | 0.395 | 0.400 | 0.333 | 0.320 | 0.373 | 0.913 | 0.914 | 0.120 |
| ShakeGestureWiimoteZ | 0.120 | 0.400 | 0.120 | 0.500 | 0.680 | 0.700 | 0.650 | 0.852 | 0.493 |
| ShapeletSim | 0.756 | 0.650 | 0.506 | 0.494 | 0.650 | 0.522 | 0.784 | 0.769 | 0.829 |
| ShapesAll | 0.773 | 0.768 | 0.058 | 0.615 | 0.720 | 0.778 | 0.925 | 0.455 | 0.046 |
| SmallKitchenAppliances | 0.707 | 0.643 | 0.336 | 0.357 | 0.480 | 0.405 | 0.101 | 0.913 | 0.914 |
| SmoothSubspace | 0.893 | 0.827 | 0.740 | 0.680 | 0.820 | 0.667 | 0.907 | 0.120 | 0.952 |
| SonyAIBORobotSurface1 | 0.712 | 0.725 | 0.696 | 0.704 | 0.622 | 0.729 | 0.650 | 0.955 | 0.830 |
| SonyAIBORobotSurface2 | 0.843 | 0.831 | 0.643 | 0.837 | 0.728 | 0.885 | 0.955 | 0.830 | 0.995 |
| Strawberry | 0.943 | 0.941 | 0.643 | 0.959 | 0.935 | 0.941 | 0.550 | 0.761 | 0.550 |
| SwedishLeaf | 0.790 | 0.792 | 0.088 | 0.701 | 0.706 | 0.830 | 0.562 | 0.626 | 0.783 |
| Symbols | 0.953 | 0.950 | 0.370 | 0.822 | 0.907 | 0.918 | 0.395 | 0.650 | 0.852 |
| SyntheticControl | 0.987 | 0.867 | 0.203 | 0.440 | 0.380 | 0.907 | 0.830 | 0.995 | 0.953 |
| ToeSegmentation1 | 0.798 | 0.772 | 0.649 | 0.645 | 0.610 | 0.737 | 0.913 | 0.808 | 0.516 |
| ToeSegmentation2 | 0.846 | 0.838 | 0.838 | 0.423 | 0.723 | 0.862 | 0.551 | 0.663 | 0.805 |
| Trace | 0.990 | 1.000 | 0.190 | 0.880 | 0.570 | 0.900 | 0.808 | 0.516 | 0.819 |
| TwoLeadECG | 0.931 | 0.904 | 0.500 | 0.969 | 0.651 | 0.848 | 0.859 | 0.174 | 0.611 |
| TwoPatterns | 1.000 | 1.000 | 0.255 | 0.496 | 0.964 | 0.562 | 0.667 | 0.562 | 0.626 |
| UMD | 0.972 | 0.993 | 0.792 | 0.806 | 0.701 | 0.854 | 0.680 | 0.766 | 0.539 |
| UWaveGestureLibraryAll | 0.916 | 0.891 | 0.138 | 0.529 | 0.948 | 0.845 | 0.974 | 0.780 | 0.928 |
| UWaveGestureLibraryX | 0.731 | 0.671 | 0.162 | 0.625 | 0.749 | 0.574 | 0.679 | 0.587 | 0.879 |
| UWaveGestureLibraryY | 0.645 | 0.606 | 0.149 | 0.439 | 0.671 | 0.523 | 0.880 | 0.679 | 0.587 |
| UWaveGestureLibraryZ | 0.659 | 0.615 | 0.129 | 0.566 | 0.658 | 0.523 | 0.805 | 0.880 | 0.679 |
| Wafer | 0.984 | 0.980 | 0.834 | 0.996 | 0.996 | 0.999 | 0.750 | 0.680 | 0.766 |
| Wine | 0.593 | 0.574 | 0.500 | 0.519 | 0.611 | 0.593 | 0.761 | 0.550 | 1.000 |
| WordSynonyms | 0.676 | 0.649 | 0.045 | 0.522 | 0.491 | 0.639 | 0.516 | 0.819 | 0.785 |
| Worms | 0.519 | 0.584 | 0.416 | 0.377 | 0.494 | 0.455 | 0.101 | 0.899 | 0.835 |
| WormsTwoClass | 0.636 | 0.623 | 0.429 | 0.571 | 0.597 | 0.610 | 0.109 | 0.101 | 0.899 |
| Yoga | 0.839 | 0.836 | 0.455 | 0.780 | 0.797 | 0.852 | 0.789 | 0.395 | 0.650 |

