# OpenReview forum: "TimePoint: Accelerated Time Series Alignment via Self-Supervised Keypoint and Descriptor Learning"
_ICML.cc/2025/Conference — ICML 2025 poster_

### Official Review · Reviewer_wuFf · 2025-03-08

**Overall Recommendation:** 3

**Summary:**

This paper proposes a keypoint and descriptor detection algorithm, called TimePoint, for time-series alignment. The authors use a deep learning approach to train a model on synthetically-generated data for detecting keypoints and descriptors. The correspondences learned by their model is passed into a DTW algorithm to perform the time-series alignment.

**Claims And Evidence:**

The paper claims that their keypoint/descriptor model trained on synthetic data yields more accurate and efficient alignment at a lower computational cost. This is because their model enables a sparser correspondence. In the experiments, they show that their method has better classification performance and runtime in terms of a kNN analysis with various baselines and the proposed method as the distance measure.

**Essential References Not Discussed:**

N/A

**Experimental Designs Or Analyses:**

Yes

**Methods And Evaluation Criteria:**

Yes, I believe the methods and evaluation criteria make sense for the problem domain.

**Other Comments Or Suggestions:**

I'm curious why the authors approach the problem from a synthetic-generation standpoint. I'm not sure if it was mentioned (at least I could not find it) for why this approach was used, as opposed to, e.g. training on real data with augmentations. I feel like this approach would be more aligned with real data and learn correspondences that might be more transferable to real-world applications. Could the authors please elaborate on this design choice, and perhaps motivate this more in their paper?

**Other Strengths And Weaknesses:**

Strengths:
- The paper is well-written
- The paper solves a clear problem and motivates their method well.
- I think the results indicate that learning data-driven, sparse correspondences leads to improved performance, and this makes sense to me.

Weaknesses:
- My main weakness is the rather limited scope of this work. It is focused on the problem of time-series alignment which is one task in a specific modality of 1-d time series, and may not be sufficiently broad enough for an ICML audience. I wonder if the authors might consider applying this sparse correspondence approach to other tasks or modalities?

**Questions For Authors:**

N/A

**Relation To Broader Scientific Literature:**

This paper builds on DTW and its extensions (e.g. softDTW, shapeDTW), which propose either different optimization strategies or handcrafted descriptors for correspondence matching. This paper is different in that it uses a deep learning strategy to learn descriptors and keypoints, where the deep learning model is trained on synthetically-generated data.

**Theoretical Claims:**

The paper doesn't have extensive proofs, but the equations in page 6 make sense to me.

---

> ### Author Rebuttal · Authors · 2025-03-31
>
> We thank the reviewer for their kind words and suggestions.
>
> # Q1 - The scope of the proposed method
>
> We appreciate the reviewer's concern regarding the scope of the proposed method. Time Series Alignment (TSA) has been a key research topic of time series analysis and machine learning in general. Machine Learning conferences such as ICML and NeurIPS publish several TSA papers at each conference. The original DTW paper has approx. 10K citations. Some key examples for papers on this topic include:
> * SoftDTW (Cuturi & Blondel, ICML 2017)
> * Drop-DTW (Neurips 2017)
> * Autowarp (NeurIPS 2018)
> * Diffeomorphic Temporal Alignments Nets (Neurips 2019)
> * Closed-form Diffeomorphic Transformations for Time Series Alignment (ICML 2022)
> * Regularization-free Diffeomorphic Temporal Alignment Nets (ICML 2023)
>
> We believe the UCR archive, that was used for evaluation and includes more than 100 real-world datasets, is diverse enough in terms of modalities (ECG, motion sensor, shape outlines, and more). Thus, the concern that the training on synthetic data won't suffice for obtaining good results on real-world datasets is unwarranted: as we showed in the paper, the approach yielded SOTA results on real-world datasets.
>
> RE: other tasks - TP could potentially be extended to other tasks such as time series classification, clustering, anomaly detection, and more. However, properly addressing these tasks is beyond the scope of the current paper. These are indeed interesting research directions that are left for future works.
>
> # Q2 - Synthetic data training vs real-world data
>
> An excellent point. We discuss this issue in lines: 065-073.
>
> We agree with the notion that training and/or fine-tuning TP on real-world data could further improve the proposed method. The reasons we chose to focus solely on synthetic data are:
> 1. We are not aware of real-world TS datasets with annotated KPs.
> 2. Using only synthetic data mitigates license-related issues that might arise.
>
> We will make this point clearer in the revised manuscript.
>
> That being said, we have conducted more experiments on real-world datasets. Specifically, as per your inquiry and other reviewers' requests, we perform a fine tuning experiment where TP is pre-trained on synthetic data and then fine-tuned on real-world data.
> Due to space limitations, they appear in our answers to other reviewers:
>
> Please see the **additional experiments**:
> 1. **Robustness to Noise** experiment - Reviewer **zrdP**
> 2. **Fine Tune on Real World Data** - Reviewer **QUAf**

---

### Official Review · Reviewer_zrdP · 2025-03-09

**Overall Recommendation:** 4

**Summary:**

The paper presents TimePoint, a self-supervised method for accelerating Dynamic Time Warping (DTW) in time series alignment by leveraging keypoint detection and descriptor learning from synthetic data. The main findings demonstrate that TimePoint significantly outperforms traditional DTW in terms of speed and accuracy, effectively addressing computational challenges and noise sensitivity in time series analysis.

**Claims And Evidence:**

The claims made in the submission regarding TimePoint's capability to enhance the speed and accuracy of Dynamic Time Warping (DTW) are well supported by extensive experimental results demonstrating significant improvements over traditional DTW methods. However, while the authors provide convincing evidence of TimePoint's performance on synthetic data, further clarification on its generalization to a wider range of real-world datasets with varying conditions may strengthen the submission's robustness.

**Essential References Not Discussed:**

No

**Experimental Designs Or Analyses:**

The experimental designs and analyses appear to be sound, particularly the ablation study which effectively demonstrates the performance improvements of the TimePoint framework across various datasets. However, further clarification on the selection criteria for datasets and the impact of synthetic data generation on real-world performance would enhance the validity of the findings.

**Methods And Evaluation Criteria:**

The proposed method, TimePoint, effectively addresses the challenges of time series alignment by utilizing a self-supervised learning approach that incorporates keypoint detection and descriptor generation, making it suitable for handling large and varied datasets. Additionally, the evaluation criteria, including extensive testing on benchmark datasets like the UCR Time Series Archive, demonstrate its robust performance and contextual relevance to real-world applications in time series analysis.

**Other Comments Or Suggestions:**

None

**Other Strengths And Weaknesses:**

The paper introduces TimePoint, a novel self-supervised framework for keypoint detection and descriptor learning specifically tailored for time series data. This adaptation from 2D keypoint detection methods to 1D signals is a significant innovation, addressing a gap in existing literature.

**Questions For Authors:**

1. How does TimePoint's performance compare to other state-of-the-art time series alignment methods in terms of robustness against noise and temporal distortions in real-world datasets?
2. What specific metrics or criteria were used to evaluate the generalization capability of TimePoint when applied to diverse real-world time series data, beyond the synthetic datasets used for training?
3. Can you elaborate on the potential limitations or challenges of using synthetic data for training TimePoint, particularly in terms of its applicability to highly variable real-world time series signals?

**Relation To Broader Scientific Literature:**

The key contributions of the paper, particularly the introduction of TimePoint for efficient time-series alignment, relate closely to existing literature in several ways:
1) Dynamic Time Warping (DTW): The paper builds on the foundational work of DTW, which has been widely used for time-series alignment due to its ability to handle elastic shifts in the temporal axis. However, it addresses DTW's limitations, such as its quadratic time complexity and sensitivity to noise, which have been noted in prior studies.
2) Keypoint Detection and Description: TimePoint adapts concepts from 2D keypoint detection methods, like SuperPoint, to the 1D time-series domain. This adaptation is significant as it addresses the unique challenges of time-series data, such as nonlinear distortions and amplitude variations, which have not been thoroughly explored in previous works.
3) Synthetic Data Generation: The introduction of SynthAlign, a synthetic dataset designed for training keypoint detection and descriptor learning, is a novel contribution. Previous research has often focused on real-world datasets, while this paper emphasizes the importance of synthetic data for self-supervised learning, a concept that has been underutilized in time-series analysis.
4) Continuous Piecewise Affine Based (CPAB) Transformations: The use of CPAB transformations to model nonlinear temporal warping is a significant advancement. While prior works have explored diffeomorphic transformations, they often lacked practical implementations for keypoint detection and descriptor extraction, which TimePoint successfully integrates.
5) Efficiency and Scalability: The paper demonstrates that applying DTW to a sparse set of keypoints and descriptors leads to substantial computational speedups and improved alignment accuracy. This finding aligns with the ongoing research in machine learning and time-series analysis that seeks to enhance efficiency without sacrificing performance, as seen in methods like FastDTW and SoftDTW.

Overall, the contributions of this paper not only advance the state of the art in time-series alignment but also provide a framework that can inspire further research in both theoretical and applied contexts within the broader scientific literature.

**Theoretical Claims:**

I reviewed the theoretical claims presented in the paper, particularly focusing on the proofs related to the efficiency of the TimePoint framework and its alignment accuracy.

---

> ### Author Rebuttal · Authors · 2025-03-31
>
> We thank the reviewer for their kind words and comprehensive response and suggestions.
>
> # Q1 - TP vs. SOTA time series alignment: robustness to noise and temporal distortions.
>
> TP was evaluated on 100+ datasets of the UCR archive. These datasets significantly vary in terms of noise and temporal distortion. In Section A. in the appendix, page 12, we show the model’s KPs and descriptors on three datasets that vary in length, frequency, domain, and distortion. Additionally, we believe the comparisons in Figure 7 and Table 1 provide a strong indication of TP robustness compared to state-of-the-art DTW-based method. While we test TP+DTW and TP+SoftDTW (Table 1), other alignment algorithms could also benefit from TP’s descriptors and KPs. Finally, we have performed a "Robustness to noise" experiment on real-world data, detailed below.
>
> # Robustness to Noise (Additional Experiment)
>
> We conducted a robustness analysis to Gaussian Blur where $\sigma\in\[0.1, 0.2\]$ and Additive Gaussian Noise (a.k.a, Jitter) where $\sigma\in\[0.1, 0.2\]$.
> Due to the random sampling of noise, we repeat the experiment 3 times. Due to the large number of experiments per dataset (noise types X noise rates X method = 2 X 4 X 3) we use a subset of 30 UCR datasets. The results are presented below:
>
>
> |                  | no_noise | Blur ($\sigma=0.1$)  |  Blur ($\sigma=0.2$) | Jitter ($\sigma=0.1$) | jitter ($\sigma=0.2$) |
> |------------------|----------|--------------|--------------|---------|---------|
> | DTW              | 0.844    | 0.843        | 0.838        | 0.801   | 0.744   |
> | TimePoint(kp=0.1)| 0.867    | 0.866        | 0.853        | 0.804   | 0.760   |
> | TimePoint(kp=0.2)| **0.881**    | **0.873**        | **0.873**        | **0.828**   | **0.791**   |
>
>
> # Q2 - What specific metrics or criteria were used to evaluate generalization?
>
> We have used the Nearest Neighbor classifier with DTW as distance measure (DTW-NN) to evaluate TP on real-world data. It is the customary benchmark for evaluating alignment algorithms (e.g., see the SoftDTW paper [Cuturi & Blondel, ICML ‘17]).
> In more detail, DTW is performed between each test and train sample. The test data is labeled according to its closest neighbor label in the train set. Higher classification accuracy indicates that the similarity measure (DTW, TP-DTW, SoftDTW, etc.) is better.
> Given that the similarity measure is fixed (DTW) compared with competing methods, the quality of the KPs and descriptors is the main variable measured for evaluation.
>
> We will clarify this point in the revised manuscript.
>
> # Q3 - Limitations using synthetic data when generalizing to real-world data
>
> Yes. As we mentioned in the limitations section (Sec. 5, Line 344), a key limitation is when the real-world data significantly differ from the synthetic data for SynthAlign. An example for such data is speech recordings. While the UCR holds some data that relates to speech recognition (see Appendix. A.1, Line 652 -Phoneme dataset), speech alignment and/or recognition is left for further research, as it requires special care (perhaps a combination of SynthAlign and text-to-speech generative model).
>
> To further answer this question, we have conducted an experiment where we fine-tune TP on real world data.
> Please see our answer to  Reviewer QUAf:
> **Fine Tune on Real World Data**.

---

### Official Review · Reviewer_qAHo · 2025-03-10

**Overall Recommendation:** 3

**Summary:**

This paper introduces TimePoint, a self-supervised framework for accelerating DTW-based time series alignment by learning keypoints and descriptors. TimePoint leverages 1D diffeomorphisms to model nonlinear temporal distortions, combined with fully convolutional and wavelet-based architectures to extract multi-scale features. Experiments demonstrate that TimePoint consistently achieves faster and more accurate alignments than standard DTW, making it a scalable solution for time-series analysis.

**Claims And Evidence:**

Yes

**Essential References Not Discussed:**

No.

**Experimental Designs Or Analyses:**

Yes

**Methods And Evaluation Criteria:**

Yes

**Other Comments Or Suggestions:**

See Weakness.

**Other Strengths And Weaknesses:**

Strengths:

1. The paper introduces an efficient time-series alignment framework which addresses the scalability limitation of traditional methods like DTW.

2. Experimentally, the proposed TP is a robust and efficient method.

Weakness:

1. Since SynthAlign relies on predefined waveforms (e.g., sine, RBF), its performance on real-world data with complex patterns (e.g., high-frequency noise, non-stationary trends) is unclear. It is recommended to conduct experiments under the real-world dataset like PTB [1].

[1] Goldberger, Ary L., et al. "PhysioBank, PhysioToolkit, and PhysioNet: components of a new research resource for complex physiologic signals." circulation 101.23 (2000): e215-e220.

2. In Figure 8, when using the full signal (L = 100%), the runtimes for DTW and TP+DTW are almost identical. This observation warrants further analysis and discussion of the results.

**Questions For Authors:**

See Weakness.

**Relation To Broader Scientific Literature:**

Not mentioned.

**Theoretical Claims:**

Yes

---

> ### Author Rebuttal · Authors · 2025-03-31
>
> We thank the reviewer for their kind words and suggestions.
> # Q1 - It is recommended to conduct experiments under the real-world dataset like PTB [1]
> Agreed, but this was already done in the paper. Please note that while TP was trained on synthetic data, all of the reported experiments were conducted on **real world datasets** of the UCR time series classification archive (i.e., section 6.2  **Classification on Real-World Data** and the results from Figures 7-8, and Tables 1-2).
>
> We evaluated TP on more than 100 datasets of various data types, such as ECG, sensors, motions (action recognition for Wii remote, smart devices), Human activity recognition (HAR) and more. The datasets were contributed by different authors.
> Specifically, the original dataset for "ECG5000", which is one those 100+ UCR datasets, is a 20-hour long ECG downloaded **from Physionet** (under the name “BIDMC Congestive Heart Failure Database”, see: https://www.timeseriesclassification.com/description.php?Dataset=ECG5000 ); i.e., the very dataset the reviewer mentioned.
>
> The UCR archive is considered a gold standard for time series alignment and is widely used for evaluating alignment algorithms such as DTW.
>
> We will make sure this is clearer in the revised manuscript.
>
> Since PhysioNet contains dozens of datasets, not all are applicable for TS alignment, we were unable to evaluate TP on further datasets (besides ECG5000) from this benchmark.
>
> ### Please also see the **additional experiments** on real world data:
> 1. **Robustness to Noise** experiment - Reviewer **zrdP**
> 2. **Fine Tune on Real World Data** - Reviewer **QUAf**
>
> # Q2 - Runtime (Figure 8)
> You are correct. The runtime for DTW and TP+DTW using 100% of KPS (the entire length of the time series) are almost identical. This is due to the low-overhead of TPs forward pass, which means that the runtime is almost entirely dominated by the time it takes to compute DTW. We will make this point clearer in the revised manuscript. Thank you for pointing it out.

---

### Official Review · Reviewer_QUAf · 2025-03-10

**Overall Recommendation:** 3

**Summary:**

This paper proposes a self-supervised method that dramatically accelerates DTW-based alignment while typically improving alignment accuracy by learning keypoints and descriptors from synthetic data.

**Claims And Evidence:**

The claims are clear and convincing except one concern.
While CPA is an effective method for modeling nonlinear temporal distortions and generating correspondences between time series signals, I am concerned that CPAB may not be sufficiently comprehensive to model all types of time series distortions, particularly those exhibiting high complexity and variability. For instance, in the case of non-stationary time series, the parameter $\theta$ in the CPAB transformation might evolve over time. I recommend that the authors conduct further studies to enhance their proposed method and ensure it can fully address all challenges posed by real-world problems.

**Essential References Not Discussed:**

No

**Experimental Designs Or Analyses:**

I recommend that the authors conduct a robustness analysis of their proposed methods. The paper utilizes CPAB warps to generate training pairs with ground-truth correspondences. However, in real-world scenarios, some pairs may be incomplete. Introducing a study where certain points are removed from the pairs and evaluating the model's performance on incomplete data could enhance its robustness to various practical scenarios.

**Methods And Evaluation Criteria:**

The paper introduces a synthetic time series dataset with known KPs and applies CPAB warps to generate training pairs with ground-truth correspondences. It further proposes a self-supervised framework, TimePoint, for detecting and describing keypoints in time series data. However, the model's performance could potentially be enhanced by fine-tuning it on real-world datasets. As noted by the authors in Section 5, TimePoint's performance may be suboptimal if the signals deviate significantly from the synthetic distribution. Therefore, I strongly recommend incorporating a fine-tuning step on real-world data to ensure robust performance and adaptability to practical scenarios.

**Other Comments Or Suggestions:**

In Section 4.2, KPs are selected by either applying a pre-defined threshold or choosing the top-K timesteps with the highest probability.
 -> In the experiments, please elaborate how sensitive the threshold to the model's performance.

**Other Strengths And Weaknesses:**

While CPA is an effective method for modeling nonlinear temporal distortions and generating correspondences between time series signals, I am concerned that CPAB may not be sufficiently comprehensive to model all types of time series distortions, particularly those exhibiting high complexity and variability. For instance, in the case of non-stationary time series, the parameter $\theta$ in the CPAB transformation might evolve over time. I recommend that the authors conduct further studies to enhance their proposed method and ensure it can fully address all challenges posed by real-world problems.

**Questions For Authors:**

While CPA is an effective method for modeling nonlinear temporal distortions and generating correspondences between time series signals, I am concerned that CPAB may not be sufficiently comprehensive to model all types of time series distortions, particularly those exhibiting high complexity and variability. For instance, in the case of non-stationary time series, the parameter $\theta$ in the CPAB transformation might evolve over time. I recommend that the authors conduct further studies to enhance their proposed method and ensure it can fully address all challenges posed by real-world problems.

The paper introduces a synthetic time series dataset with known KPs and applies CPAB warps to generate training pairs with ground-truth correspondences. It further proposes a self-supervised framework, TimePoint, for detecting and describing keypoints in time series data. However, the model's performance could potentially be enhanced by fine-tuning it on real-world datasets. As noted by the authors in Section 5, TimePoint's performance may be suboptimal if the signals deviate significantly from the synthetic distribution. Therefore, I strongly recommend incorporating a fine-tuning step on real-world data to ensure robust performance and adaptability to practical scenarios.

 I recommend that the authors conduct a robustness analysis of their proposed methods. The paper utilizes CPAB warps to generate training pairs with ground-truth correspondences. However, in real-world scenarios, some pairs may be incomplete. Introducing a study where certain points are removed from the pairs and evaluating the model's performance on incomplete data could enhance its robustness to various practical scenarios.

n Section 4.2, KPs are selected by either applying a pre-defined threshold or choosing the top-K timesteps with the highest probability.
 -> In the experiments, please elaborate how sensitive the threshold to the model's performance.

**Relation To Broader Scientific Literature:**

The efficient and scalable alignment of time series is a critical research topic within the time series community. This study introduces a self-supervised method that significantly accelerates DTW-based alignment, offering key advancements and valuable insights to the field.

**Theoretical Claims:**

The proofs for theoretical claims are correct.

---

> ### Author Rebuttal · Authors · 2025-03-31
>
> We thank the reviewer for their kind words and comprehensive response and suggestions.
>
> # Q1 - Non-stationary time series  (NSTS)
> This is an excellent point. A single CPAB warp implies a stationary velocity field, while the CPAB prior restricts the warps to avoid unrealistic distortions. Three factors affect the alignment quality:
> 1. KP detection is unaffected by NSTS as it is performed independently for each TS.
> 2. Descriptors are also computed independently. Since each descriptor's receptive field is quite large, it might be affected by NSTS.
> 3. The matching is computed using DTW on TP’s features at KPs locations. DTW is not restricted to a specific type of misalignment between TS, and in TP’s case only relies on the features' similarity.
>
> Empirically, the UCR archive consists of more than a 100 of real-world datasets, including several NSTS datasets. E.g., “ECGFiveDays” dataset holds ECGs that were recorded five days apart. TP reaches 94.2% test accuracy while the best competitor achieves 92% (see Tables E2 and E3 at our appendix).
>
> As an aside, one can compose CPAB warps, and, partially because CPAB warps are not closed under composition, this has the effect of integrating a non-stationary velocity field. However, in 1D, this effect can be well approximated by taking a finer tessellation. Either way, we haven't noticed an empirical need to do so.
> # Q2 - Fine-tuning on real-world data
> True, the performance could potentially be enhanced by fine-tuning the model on real-world data. The reasons we chose to focus the training on synthetic data (as opposed to the test, that was done on real-world data) are:
> 1. We are unaware of real-world TS dataset with enough annotated KPs.
> 2. Using synthetic data bypasses any license-related issues that might arise.
>
> That said, and in accordance with the reviewers' suggestions, here in the rebuttal we include additional results, obtained after fine-tuning TP on real-world data (see below).
>
> # Fine-Tune on Real-World Data
> We thank all reviewers for their suggestions and inquiries regarding fine-tuning TP on real-world data.
> The experiment details are as follows:
> * We used the same method to compute KP as we did for SynthAlign (local minima/maxima, etc.).
> * We fine-tuned TP after it was first trained on SynthAlign.
> * The training data contained virtually the entire UCR archive (a few datasets were omitted due to technical reasons).
> * We created pairs with known correspondences by applying two CPAB warps to each input signal (i.e., each signal is augmented twice, and those augmented views are the input for TP and loss functions).
> * Training for 2K epochs
>
> Below we report results for 10%/20%/50% of the length (the experiment
> with 100\% didn't finish in time for the rebuttal, but we'll include it in the revised manuscripts).
>
> **Results**
> | Method                   | Baseline (DTW) | 10%   | 20%   | 50%   |
> |--------------------------|----------------|-------|-------|-------|
> | TP (Synth)               | 0.706          | 0.707 | 0.721 | 0.710 |
> | TP (Synth + Fine Tune)   | 0.706          | **0.777** | **0.790** | **0.769** |
>
> The results show that fine-tuning TP on real-world datasets yields a significant improvement in performance across all KPs percentages. The table follows Table 1 from the paper, where the results in the first row are taken from the paper and the second row from this fine-tune experiment. The runtime remains the same as in Table 1.
>
> We thank the reviewers for suggesting this experiment. Fine-tuning TP on real-world data, using our proposed training scheme, is a useful addition to the overall method.  We will incorporate this experiment in the revised manuscript.
>
> # Q3 - Robustness with missing KPs
> Re missing data, we agree such analysis can be benficial. That said, we are unsure exactly what the reviewer meant: “[...] where certain points are removed from the pairs and evaluating on incomplete data“. Since during inference KPs are computed automatically by TP, the scenario where points are removed never occurs in our current setting.
>
> We have conducted a **“Robustness to Noise”** experiment using Blurring and Jitter. (please see our answer to **zrdP**).
>
> # Q4 - Sensitivity to the threshold.
> The choice of threshold is critical for TP’s performance. To that end, Table 1 shows the performance for 10\%, 20\%, 50\%, 100\% of KPs. Effectively, each value corresponds to a different threshold (the reported runtime includes sorting KPs by their probability, so the overhead is minimal). Since the evaluation was conducted on 100+ datasets, choosing a fixed threshold for all dataset will be sub-optimal. Thus, we believe this adaptive strategy is a good heuristic when working with a large number of datasets. When working on a specific dataset, cross-validation may also be used for choosing the value.
> We will elaborate on the model’s sensitivity to the choice of threshold in the revised manuscript.

---

### Decision · Program_Chairs · 2025-05-01

**Decision:**

Accept (poster)

**Comment:**

This paper proposes TimePoint, a self-supervised method for accelerating DTW-based time series alignment by learning keypoints and descriptors from synthetic data. The approach is well-motivated, technically sound, and addresses the high computational cost of traditional DTW.

Reviewers highlight strong performance across over 100 real-world datasets, solid theoretical grounding, and a clear problem setup. Concerns about generalization, robustness to non-stationary signals, and reliance on synthetic data are valid but addressed effectively in the rebuttal. The authors provide new fine-tuning experiments, noise robustness results, and clarify the diversity of evaluation datasets.

While the scope is focused, the contribution is significant and addresses a core challenge in time-series analysis. The method is likely to be useful in a range of downstream tasks.

The paper presents a practical, validated contribution to scalable time series alignment.